# Near-infrared spectroscopy estimation of combined skeletal muscle oxidative capacity and O$_2$ diffusion capacity in humans

Andrea M. Pilotto[1,2] (ID), Alessandra Adami[3] (ID), Raffaele Mazzolari[2,4] (ID), Lorenza Brocca[2] (ID), Emanuela Crea[2], Lucrezia Zuccarelli[1] (ID), Maria A. Pellegrino[2,5] (ID), Roberto Bottinelli[2,5] (ID), Bruno Grassi[1] (ID), Harry B. Rossiter[6] (ID) and Simone Porcelli[2,7] (ID)

[1] *Department of Medicine, University of Udine, Udine, Italy*

[2] *Department of Molecular Medicine, Institute of PhysiologyUniversity of Pavia, Pavia, Italy*

[3] *Department of Kinesiology, University of Rhode Island, Kingston, RI, USA*

[4] *Department of Physical Education and Sport, University of the Basque Country (UPV/EHU), Vitoria-Gasteiz, Spain*

[5] *Interdipartimental Centre for Biology and Sport Medicine, University of Pavia, Pavia, Italy*

[6] *Division of Respiratory and Critical Care Physiology and Medicine, The Lundquist Institute for Biomedical Innovation at Harbor–UCLA Medical Center, Torrance, CA, USA*

[7] *Institute of Biomedical Technologies, National Research Council, Milan, Italy*

Handling Editors: Michael Hogan & Ross Pollock

The peer review history is available in the Supporting Information section of this article (https://doi.org/10.1113/JP283267#support-information-section).

**Abstract** The final steps of the O$_2$ cascade during exercise depend on the product of the microvascular-to-intramyocyte $P_{O_2}$ difference and muscle O$_2$ diffusing capacity ($Dm_{O_2}$). Non-invasive methods to determine $Dm_{O_2}$ in humans are currently unavailable. Muscle oxygen

The Journal of Physiology

uptake ($m\dot{V}_{O_2}$) recovery rate constant ($k$), measured by near-infrared spectroscopy (NIRS) using intermittent arterial occlusions, is associated with muscle oxidative capacity *in vivo*. We reasoned that $k$ would be limited by $Dm_{O_2}$ when muscle oxygenation is low ($k_{LOW}$), and hypothesized that: (i) $k$ in well oxygenated muscle ($k_{HIGH}$) is associated with maximal $O_2$ flux in fibre bundles; and (ii) $\Delta k$ ($k_{HIGH} - k_{LOW}$) is associated with capillary density (CD). Vastus lateralis $k$ was measured in 12 participants using NIRS after moderate exercise. The timing and duration of arterial occlusions were manipulated to maintain tissue saturation index within a 10% range either below (LOW) or above (HIGH) half-maximal desaturation, assessed during sustained arterial occlusion. Maximal $O_2$ flux in phosphorylating state was $37.7 \pm 10.6$ pmol s$^{-1}$ mg$^{-1}$ ($\sim 5.8$ ml min$^{-1}$ 100 g$^{-1}$). CD ranged 348 to 586 mm$^{-2}$. $k_{HIGH}$ was greater than $k_{LOW}$ ($3.15 \pm 0.45$ *vs.* $1.56 \pm 0.79$ min$^{-1}$, $P < 0.001$). Maximal $O_2$ flux was correlated with $k_{HIGH}$ ($r = 0.80$, $P = 0.002$) but not $k_{LOW}$ ($r = -0.10$, $P = 0.755$). $\Delta k$ ranged $-0.26$ to $-2.55$ min$^{-1}$, and correlated with CD ($r = -0.68$, $P = 0.015$). $m\dot{V}_{O_2}$ $k$ reflects muscle oxidative capacity only in well oxygenated muscle. $\Delta k$, the difference in $k$ between well and poorly oxygenated muscle, was associated with CD, a mediator of $Dm_{O_2}$. Assessment of muscle $k$ and $\Delta k$ using NIRS provides a non-invasive window on muscle oxidative and $O_2$ diffusing capacity.

(Received 28 April 2022; accepted after revision 27 July 2022; first published online 5 August 2022)

**Corresponding author** S. Porcelli, Department of Molecular Medicine, Institute of Physiology, University of Pavia, 27100 Pavia, Italy. Email: simone.porcelli@unipv.it

**Abstract figure legend** A NIRS probe was positioned on vastus lateralis muscle and used to assess both muscle oxidative and oxygen diffusing capacity *in vivo* by intermittent arterial occlusions. Timing and duration of each occlusion were manipulated to maintain tissue saturation index within a 10 percentage range either below (LOW) or above (HIGH) half-maximal desaturation. Recovery rate constant ($k$) of muscle oxygen consumption ($m\dot{V}_{O_2}$) obtained by NIRS was correlated with phosphorylating oxidative capacity of permeabilized muscle fibre bundles. Changes in recovery rate constant ($k$) were correlated with capillary density calculated from cross-section of muscle samples obtained by immuno-fluorescence.

## Key points

- We determined post-exercise recovery kinetics of quadriceps muscle oxygen uptake ($m\dot{V}_{O_2}$) measured by near-infrared spectroscopy (NIRS) in humans under conditions of both non-limiting (HIGH) and limiting (LOW) $O_2$ availability, for comparison with biopsy variables.
- The $m\dot{V}_{O_2}$ recovery rate constant in HIGH $O_2$ availability was hypothesized to reflect muscle oxidative capacity ($k_{HIGH}$) and the difference in $k$ between HIGH and LOW $O_2$ availability ($\Delta k$) was hypothesized to reflect muscle $O_2$ diffusing capacity.
- $k_{HIGH}$ was correlated with phosphorylating oxidative capacity of permeabilized muscle fibre bundles ($r = 0.80$).
- $\Delta k$ was negatively correlated with capillary density ($r = -0.68$) of biopsy samples.
- NIRS provides non-invasive means of assessing both muscle oxidative and oxygen diffusing capacity *in vivo*.

**Andrea M. Pilotto** is a Research Assistant at the Department of Molecular Medicine, University of Pavia, and a PhD student in Biomedical Sciences and Biotechnology at the Department of Medicine, University of Udine. He completed MS Degree in Sport Science at the University of Milan. His research focuses on molecular, metabolic and physiological mechanisms underlying adaptations to exercise, with particular emphasis on the effects of exercise training on mitochondrial adaptations.

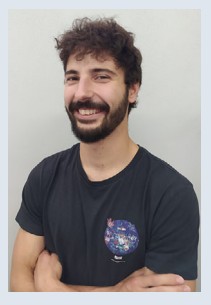

## Introduction

The primary source of ATP supply in skeletal muscle during endurance exercise is ADP phosphorylation coupled to the reduction of $O_2$ (Picard et al., 2016). Two primary resistances, limiting the maximal conductance of $O_2$ from the atmosphere to the muscle mitochondrion, reside within the cardiovascular system (i.e. convective $O_2$ transport) and at the muscle capillary–myocyte interface (i.e. diffusive $O_2$ transport) (Richardson, Knight et al., 1995; Wagner, 1992, 1995, 2000).

The interaction of the maximum rate of convective $O_2$ transport and muscle $O_2$ diffusing capacity ($Dm_{O_2}$) determines the maximal muscle $O_2$ uptake ($m\dot{V}_{O_2}$) (Roca et al., 1992), the final steps of the $O_2$ cascade determining $m\dot{V}_{O_2}$ depend on the product of the transmembrane $P_{O_2}$ gradient [microvascular (mv) to intramyocyte (im) $P_{O_2}$] and the muscle diffusing capacity for $O_2$ ($Dm_{O_2}$) (Fick's law of diffusion):

$$m\dot{V}O_2 = Dm_{O_2} \times \left(Pmv_{O_2} - Pim_{O_2}\right) \qquad (1)$$

where ($Pmv_{O_2}$ – $Pim_{O_2}$) is strongly dependent on convective $O_2$ delivery and muscle $O_2$ demand (as a function of power output). This results in $Pim_{O_2}$ becoming essentially constant at ~1–5 mmHg during exercise above ~50% maximum $O_2$ uptake ($\dot{V}_{O_2 max}$) (Clanton et al., 2013; Richardson, Noyszewski, Kendrick, Leigh & Wagner, 1995). $Dm_{O_2}$, on the other hand, is a complex function of muscle capillarity, the surface area of apposition of red blood cells to capillary endothelium, red blood cell capillary transit time, haemoglobin volume, and the $O_2$ solubility properties within the diffusion pathway (Bebout et al., 1993; Groebe & Thews, 1986; Honig et al., 1984; Poole et al., 2020; Wagner, 1995). Notwithstanding these confounding variables, $Dm_{O_2}$ is related to capillary density [CD; the number of capillaries per summed muscle fibre cross-sectional area (CSA)] (Hepple et al., 2000; Poole et al., 2020, 2021, 2022; Saltin & Gollnick, 1983). The observed linear relationship between estimated $Pmv_{O_2}$ and $\dot{V}_{O_2 max}$ supports the concept that $Dm_{O_2}$ is a major limiting variable for $m\dot{V}_{O_2 max}$ in humans (Richardson et al., 1999; Roca et al., 1992). Methods to estimate $Dm_{O_2}$ in humans *in vivo* are complex and previous attempts on quadriceps muscle required invasive procedures with repeated exercise tests using breathing of gas mixtures containing high and low fractions of inspired $O_2$. Another approach, using non-invasive venous occlusion plethysmography to assess capillarity filtration (Brown et al., 2001), provides an indirect estimation of limb capillarity, but is sensitive to changes in oncotic pressure, endothelial tight junctions and influenced by all tissues within the limb, not only muscle (Hunt et al., 2013).

We sought to simplify assessment of $Dm_{O_2}$ in human quadriceps using a more muscle specific approach. We reasoned that, if we could make non-invasive measurements of $m\dot{V}_{O_2}$ under two different ($Pmv_{O_2}$ – $Pim_{O_2}$) conditions (i.e. HIGH and LOW), we would be able to solve for $Dm_{O_2}$ by simultaneous subtraction of the two unknowns ($m\dot{V}_{O_2}$ and $Pmv_{O_2}$, respectively) in eqn (1), assuming, as in Roca et al. (1992), that $Pim_{O_2}$ is negligible. Near-infrared spectroscopy (NIRS) provides a relatively simple and non-invasive means to estimate muscle oxidative capacity (Grassi & Quaresima, 2016; Hamaoka & McCully, 2019). The recovery rate constant of $m\dot{V}_{O_2}$ ($k$) established from the rate of decline in muscle tissue saturation index (TSI) under serial, intermittent, arterial occlusions (Adami & Rossiter, 2018; Adami et al., 2017; Hamaoka et al., 1996; Motobe et al., 2004; Ryan et al., 2012) shows good agreement with estimates of muscle oxidative capacity by other techniques, such as the phosphocreatine recovery time constant ($r = 0.88$–$0.95$) (Ryan et al., 2013) or respiratory rates in fibre bundles ($r = 0.61$–$0.74$) (Ryan et al., 2014).

We modified the NIRS-based assessment of $m\dot{V}_{O_2}$ by manipulating the timing and duration of the intermittent arterial occlusions, thereby controlling the mean ($Pmv_{O_2}$ – $Pim_{O_2}$) at, separately, both HIGH (non-$O_2$ limiting) and LOW ($O_2$ limiting) values following moderate exercise. More specifically, we used $k$ as proxy for $m\dot{V}_{O_2}$ and TSI as a proxy of $Pmv_{O_2}$. Although not without limitations, this approach led us to solve the Fick's law of diffusion as:

$$k = Dm_{O_2} \times \text{TSI}$$

where $Pim_{O_2}$ is considered negligible.

Then, we calculated $k$ for both HIGH ($k_{HIGH}$) and low ($k_{LOW}$) TSI values and we compared the $k$ values with variables obtained from biopsy taken from the same muscle location and individuals. We hypothesized that the recovery rate constant of $m\dot{V}_{O_2}$ in high TSI conditions ($k_{HIGH}$) is associated with muscle oxidative capacity assessed using high resolution respirometry of permeabilized muscle fibre bundles.

Finally, we estimated $Dm_{O_2}$ from the difference in $k$ values obtained in HIGH and LOW conditions ($\Delta k = k_{HIGH} - k_{LOW}$) and by calculating $\Delta$TSI, according to:

$$\Delta k / \Delta \text{TSI} = Dm_{O_2} \sim \text{CD}$$

We then tested the hypothesis that the difference in the recovery rate constant of $m\dot{V}_{O_2}$ between HIGH and LOW TSI conditions ($\Delta k$) is associated with CD from biopsy histology.

## Methods

### Subjects

Twelve moderately trained male ($n = 7$) and female ($n = 5$) adult participants (age $28 \pm 5$ years; weight

64.3 ± 10.2 kg; height 173 ± 7 cm) were recruited from the local community. All participants completed a health history questionnaire to ensure there was no presence of chronic disease. None of the participants took any medications known to alter metabolism. Participants were fully informed about the aims, methods and risks, and gave their written informed consent prior to enrollment. All procedures were in accordance with the *Declaration of Helsinki* and the study was approved by the local ethics committee (Besta 64-19/07/2019).

### Study design

Participants visited the laboratory on four non-consecutive days over a 2 week period (Fig. 1). They were instructed to abstain from strenuous physical activity for at least 24 h prior to each testing session (48 h for the biopsy trial). At visit 1, anthropometric measurements were taken, and an incremental cardiopulmonary exercise test to the limit of tolerance was administered on an electronically braked cycle ergometer (LC-6; Monark, Vansbro, Sweden) to determine $\dot{V}_{O_2peak}$ and gas exchange threshold (GET). At visits 2 and 3, participants performed repeated muscle oxidative capacity tests within HIGH or LOW muscle oxygenation conditions (two repeats at each visit), immediately after 5 min constant work-rate cycling at 80% GET (Zuccarelli et al., 2020). At visit 4, ∼100 mg of skeletal muscle was obtained from the *vastus lateralis* muscle by percutaneous conchotome muscle biopsy under local anaesthesia (1% lidocaine) for muscle respirometry and morphology.

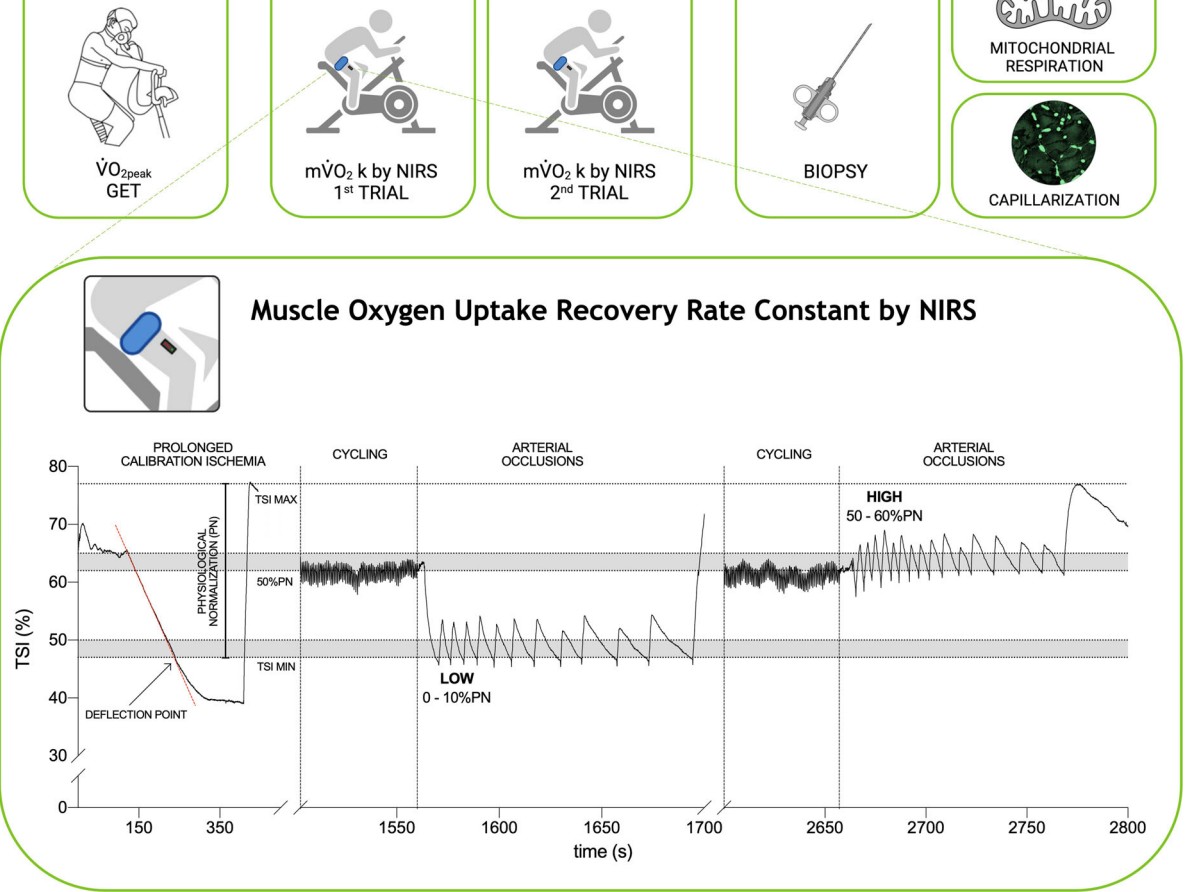

**Figure 1. Study design and muscle recovery rate constant (*k*) protocol by NIRS**
Participants visited the laboratory on four occasions. Visit 1 was used to determine peak oxygen uptake ($\dot{V}_{O_2peak}$) and gas exchange threshold (GET). Visits 2 and 3 were used to determine the physiological normalization (PN) of quadriceps TSI following sustained arterial occlusion, and the muscle $\dot{V}_{O_2}$ recovery rate constant (*k*) in well oxygenated (HIGH) and poorly oxygenated (LOW) conditions. Two *k* measurements were performed at each visit. To measure *k*, participants initially cycled for 5 min at 80% GET, followed immediately by 10–20 intermittent arterial occlusions (300 mmHg). The duration and timing of repeated occlusions were modulated to maintain TSI in two different ranges: from 0% to 10% of PN (LOW) and from 50% to 60% of PN (HIGH). A muscle biopsy was obtained during visit 4.

## Incremental exercise

Power during step-incremental cycling was increased 10 to 15 W every minute, depending on the individual's fitness. Participants were instructed to maintain constant cadence at their preferred value (between 70 and 85 rpm). Intolerance was defined when participants could no longer maintain their chosen pedalling frequency despite verbal encouragement. Pulmonary gas exchange and ventilatory variables were determined breath-by-breath using a metabolic cart (Vyntus CPX; Vyaire Medical GmbH, Höchberg, Germany), which was calibrated in accordance with the manufacturer's instructions before each test. Heart rate was recorded using a chest band (HRM-Dual; Garmin, Olathe, KS, USA) and rating of perceived exertion (RPE) was determined using Borg scale of 6–20 (Borg, 1982). At rest, and at 1, 3 and 5 min of recovery, 20 μL of capillary blood was obtained from a preheated earlobe for blood lactate concentration (Biosen C-line; EKF, Hamm, Germany). Peak cardiopulmonary variables were measured from the highest 20 s mean values prior to intolerance. GET was determined by two independent investigators by using the modified 'V slope' method (Beaver et al., 1986). The power at GET was estimated after accounting for the individual's $\dot{V}_{O_2}$ mean response time (Whipp et al., 1981).

## Muscle oxygen uptake recovery rate constant by NIRS

The m$\dot{V}_{O_2}$ recovery $k$ was measured using an approach modified from Zuccarelli et al. (2020). Oxygenation changes of the vastus lateralis were sampled at 10 Hz by a wireless, portable, continuous-wave, spatially resolved, NIRS device (PortaLite; Artinis, Elst, The Netherlands). Briefly, this device is equipped with three fibre optic bundles: NIR light is emitted from three optodes at two wavelengths (760 and 850 nm) and received from a fourth optode for transmission back to the data acquisition unit to determine the relative concentrations of deoxygenated and oxygenated heme groups contained in haemoglobin (Hb) and myoglobin (Mb). This method does not distinguish between the contributions of Hb and Mb to the NIRS signal, but Mb signal was assumed to be of minor impact compared to the contribution of Hb (Grassi & Quaresima, 2016). Relative concentrations of deoxy-(haemoglobin + myoglobin) ($\Delta$[deoxy(Hb + Mb)]) and oxy-(haemoglobin + myoglobin) ($\Delta$[oxy(Hb + Mb)] were measured in the tissues ~1.5–2 cm beneath the probe, with respect to an initial value obtained at rest before any procedure arbitrarily set equal to zero. From these measurements, relative changes in total haemoglobin and myoglobin ($\Delta$[tot(Hb + Mb)] = $\Delta$[oxy(Hb + Mb)] + $\Delta$[deoxy(Hb + Mb)]) and the Hb difference ($\Delta$[diff(Hb + Mb)] = $\Delta$[oxy(Hb + Mb)] – $\Delta$[deoxy(Hb + Mb)]) were calculated. In addition, the TSI (%) was measured using the spatially resolved spectroscopyapproach (Ferrari et al., 2004).

The skin at the NIRS probe site was shaved before the probe was placed longitudinally on the lower third of vastus lateralis muscle (~10 cm above the knee joint), and secured with a black patch and elastic bandage. The location of the probe was marked using a skin marker to ensure the placement location was similar across all visits. The mean thickness of the skin and subcutaneous tissue at the NIRS probe site (7.8 ± 3.1 mm) was measured using a skinfold caliper (Holtain Ltd, Crymych, UK). A 13 × 85 cm rapid-inflation pressure-cuff (SC12D; Hokanson, Bellevue, WA, USA) was placed proximally on the same thigh and attached to an electronically controlled rapid cuff-inflator (E20; Hokanson).

When participants were seated on a cycle ergometer, baseline TSI and $\Delta$[tot(Hb + Mb)] were measured over 2 min of rest. Subsequently, a prolonged arterial occlusion (300 mmHg) was performed until TSI plateaued (typically ~120 s). The cuff was instantly deflated and muscle reoxygenation was recorded until a steady-state was reached (typically ~3 min). This procedure identified the physiological normalization (PN) of TSI which was standardized to 0% at the deflection point (TSI min) and 100% at the maximum value reached during reperfusion (TSI max) (Adami et al., 2017) (Fig. 1). Participants then cycled for 5 min at a target of 80% GET, followed by an immediate stop and 10–20 intermittent arterial occlusions at 300 mmHg. Duration and timing of the repeated occlusions were controlled by the investigator to maintain TSI in two different ranges: from 0% to 10% of PN (LOW) and from 50% to 60% of PN (HIGH), where the total amplitude of PN was used as 0% to 100% reference range (Fig. 1). The HIGH range was selected to ensure that occlusions were performed under well oxygenated conditions, and to avoid the reduction in $P_{O_2}$ could limit m$\dot{V}_{O_2}$ (i.e. maintaining TSI above 50% of the physiological normalization) (Adami & Rossiter, 2018; Haseler et al., 2004). The LOW range was selected as the lowest boundary to evaluate m$\dot{V}_{O_2}$ recovery $k$ in poorly oxygenated conditions, without overstepping the deflection point (i.e. where TSI during occlusions loses linearity). On the same day, two repetitions of repeated occlusions protocol, separated by a resting period (typically ~5 min), were performed in a randomized order for both the LOW and HIGH experimental conditions.

The rate of muscle desaturation during each intermittent arterial occlusion (TSI, % s$^{-1}$) was fitted to estimate the exponential m$\dot{V}_{O_2}$ recovery or $k$, as described previously (Adami et al., 2017) (Fig. 2). Data were quality checked before curve fitting to remove invalid values or outliers i.e. low initial TSI values, or incomplete occlusions (Beever et al., 2020). The test–retest variability of $k$ was first assessed. Subsequently $k$ within each condition ($k_{HIGH}$ and $k_{LOW}$) was calculated from the mean of the

two repeated measurements, and the difference between these conditions was calculated ($\Delta k = k_{HIGH.} - k_{LOW}$). $\Delta[\text{tot}(Hb + Mb)]$ above rest was measured during the arterial occlusions of both HIGH and LOW conditions.

## Muscle biopsy

Resting muscle biopsies were taken from the vastus lateralis muscle using a 130 mm Weil-Blakesley rongeur (NDB-2; Fehling Instruments, GmbH&Co, Karlstein am Main, Germany) under local anaesthesia (1% lidocaine). After collection, muscle samples were cleaned of excess blood, fat and connective tissue in ice-cold BioPS, a biopsy-preserving solution containing (in mM) 2.77 CaK$_2$EGTA, 7.23 K$_2$EGTA, 5.77 Na$_2$ATP, 6.56 MgCl$_2$, 20 taurine, 50 Mes, 15 Na$_2$phosphocreatine, 20 imidazole and 0.5 dithiothreitol adjusted to pH 7.1 (Doerrier et al., 2018). A portion of each muscle sample ($\sim$10–20 mg) was immediately placed in BioPS plus 10% (w/v) fatty acid free bovine serum albumin (BSA) and 30% (v/v) DMSO

and quickly frozen in liquid nitrogen; subsequently this portion was stored at $-80°C$ for measurements of mitochondrial respiration within 1 month (Kuznetsov et al., 2003). The remaining portion were fixed in OTC (Tissue-Tek; Sakura Finetek Europe, Zoeterwoude, The Netherlands) embedding medium, frozen in N$_2$-cooled isopentane and stored at $-80°C$ for subsequent histology.

## Preparation of permeabilized fibres

Fibre bundles were quickly thawed at room temperature by immersion in BioPS containing 2 mg mL$^{-1}$ BSA to remove any residual DMSO from the tissue (Kuznetsov et al., 2003; Wüst et al., 2012). Fibes were mechanically separated with pointed forceps in ice-cold BioPS under magnification (70×) (Stereomicroscope CRYSTAL-PRO; Konus-Optical and Sports Systems, Verona, Italy). The plasma membrane was permeabilized by gentle agitation for 30 min at 4°C in 2 mL of BioPS containing 50 µg mL$^{-1}$ saponin, washed for 10 min in 2 mL of MiR06

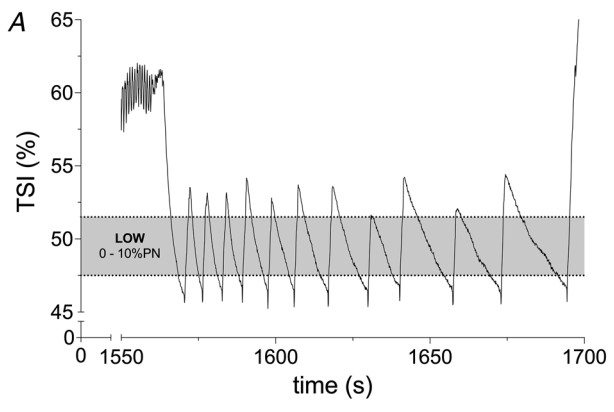

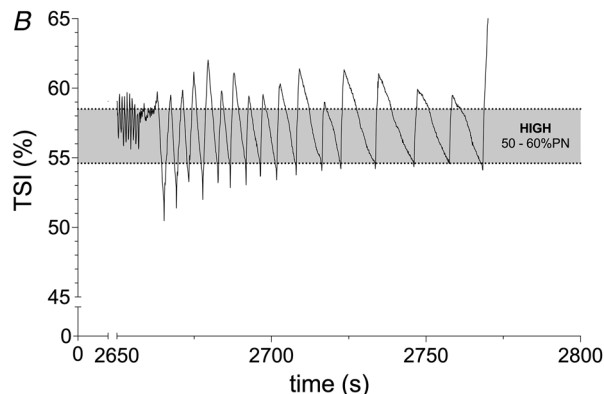

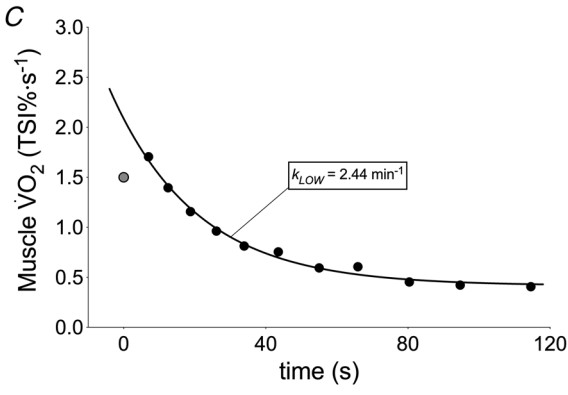

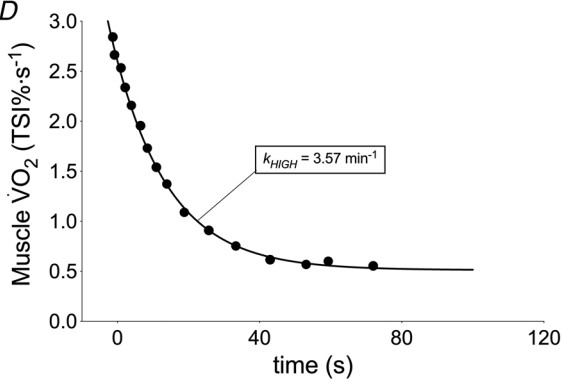

**Figure 2. Representative tissue saturation index (TSI) responses during repeated arterial occlusions of the quadriceps following moderate exercise in LOW and HIGH conditions**
Representative muscle TSI profiles and m$\dot{V}_{O_2}$ recovery kinetics during intermittent arterial occlusions following 5 min moderate intensity cycling. *A*, TSI profile in poorly oxygenated (LOW; TSI = 0–10% of physiological normalization) condition. *B*, TSI profile in well oxygenated (TSI = 50–60% of physiological normalization) condition. *C*, m$\dot{V}_{O_2}$ recovery and exponential fit (black line) for calculation of $k_{LOW}$. *D*, m$\dot{V}_{O_2}$ recovery and exponential fit (black line) for calculation of $k_{HIGH}$. Grey point represents outlier, excluded from the analysis (see Methods). *k* is the recovery rate constant (*n* = 1).

(MiR05 + catalase 280 IUmL$^{-1}$) (in mM, unless specified) 0.5 EGTA, 3 MgCl$_2$, 60 potassium lactobionate, 20 taurine, 10 KH$_2$PO$_4$, 20 Hepes, 110 sucrose and 1 g L$^{-1}$ BSA, essentially fatty acid–free (pH 7.1) (Doerrier et al., 2018) and blotted prior to being weighed.

## Mitochondrial respiration

Mitochondrial respiration was measured in duplicate or triplicate, using 3–6 mg of wet weight of muscle fibres, in 2 mL of MiR06 at 37°C containing myosin II-ATPase inhibitor (25 μM blebbistatin dissolved in DMSO, 5 mM stock) to inhibit contraction (Perry et al., 2011) (Oxygraph-2k; Oroboros, Innsbruck, Austria). Chamber O$_2$ concentration was maintained between 250 and 450 nmol mL$^{-1}$ (average O$_2$ partial pressure 250 mmHg) to avoid O$_2$ limitation of respiration. Intermittent reoxygenation steps were performed during the experiments by injections of 1 to 3 μL of 0.3 mM H$_2$O$_2$, which was instantaneously dismutated by catalase, already present in the medium, to O$_2$ and H$_2$O. Instruments were calibrated in accordance with the manufacturer's instructions (Pesta & Gnaiger, 2012).

A substrate–uncoupler–inhibitor titration protocol was used (Doerrier et al., 2018; Salvadego et al., 2016, 2018) in the order: glutamate (10 mM) and malate (4 mM) was added to assess LEAK respiration through complex I (CI$_L$). ADP (10 mM) was added to assess maximal oxidative phosphorylation (OXPHOS) capacity through CI (CI$_P$). Succinate (10 mM) was added to assess OXPHOS capacity through CI + complex II (CI + II$_P$). Cytochrome *c* (10 μM) added to test for outer mitochondrial membrane integrity. Stepwise additions of carbonyl cyanide *p*-trifluoromethoxyphenylhydrazone (0.5–1.5 μM) were used to measure electron transport system capacity through CI + II (CI + II$_E$). Inhibition of CI by rotenone (1 μM) determined electron transport system capacity through CII (CII$_E$), wheras addition of antimycin A (2.5 μM) was used to measure residual oxygen consumption, which was subtracted from all measurements. Results were expressed in pmol s$^{-1}$ mg$^{-1}$ wet weight calculating mean values of duplicate analyses. At the conclusion of each experiment, muscle samples were removed from the chamber, immediately frozen in liquid nitrogen and then stored at −80°C until measurement of CS activity.

## Citrate synthase activity

Muscle samples were thawed and underwent a motor-driven homogenization in a precooled 1 mL glass-glass potter (DWK Lifescience, Mainz, Germany). The muscle specimen was suspended 1:50 w/v in a homogenization buffer containing sucrose (250 mM),

Tris (20 mM), KCl (40 mM) and EGTA (2 mM) with 1:50 v/v protease (P8340; Sigma, St Louis, MO, USA) inhibitors. The specimen was homogenized in an ice-bath with 20 strokes at 500 rpm; before the last hit Triton X-100 (0.1% v/v) was added to the solution. After this, the sample was left in ice for 30 min. The homogenate was centrifuged at 14 000 *g* for 10 min. The supernatant was used to evaluate protein concentration according to the method of Lowry et al. (1951). Protein extracts (5−10−15 μg) were added to each well of a 96 well microplate along with 100 μL of 200 mM Tris, 20 μL of 1 mM 5,5′-dithiobis-2-nitrobenzoate, freshly prepared, 6 μL of 10 mM acetyl-coenzyme A and mQ water to a final volume of 190 μL. A background ΔAbs, to detect any endogenous activity by acetylase enzymes, was recorded for 90 s with 10 s intervals at 412 nm at 25°C using an EnSpire 2300 Multilabel Reader (PerkinElmer, Waltham, MA, USA). The ΔAbs was subtracted from the one given after the addition of 10 μL of 10 mM oxalacetic acid that started the reaction. All assays were performed at 25°C in triplicate on homogenates. Activity was expressed as nmol min$^{-1}$ (mU) per mg of protein. This protocol was modified from (Spinazzi et al., 2012; Srere, 1969).

## CSA analysis

Muscle fibre CSA was determined from several transverse sections (10 μm thick) obtained from muscle samples and probed with anti-dystrophin antibody. Fluorescence images were visualized by microscopy (U-CMAD3; Olympus, Tokyo, Japan). Fibre CSA was measured with ImageJ (NIH, Bethesda, MD, USA) and expressed in square micrometres. In total, 125–150 fibres per sample were measured.

## Capillarization

Several transverse 10 μm sections were obtained from muscle samples mounted in OTC. Sections were collected at –20 to 22°C on the surface of a polarized glass slide. Cryosections were fixed with methanol in ice for 15 min, washed three times (5 min each) in PBS (NaCl 136 mM, KCl 2 mM, Na$_2$HPO$_4$ 6 mM, KH$_2$PO$_4$ 1 mM) at room temperature (RT) and incubated in 1% Triton X-100 in PBS for 30 min (RT). Cryosections were then incubated with blocking reagents (4% BSA in 1% Triton X-100 in PBS + 5% goat serum) for 30 min (RT), raised with PBS (three times for 5 min each) and probed with anti-CD31 (dilution 1:100; Abcam, Cambridge, UK) overnight at 4°C. After three washes (5 min each) in PBS, cryosections were incubated with Alexa-Fluor 488 anti-mouse (dilution 1:200; Abcam) for 60 min at room temperature. Finally, the samples were probed with anti-dystrophin (dilution 1:500 dilution) for 60 min at room temperature and then

with Alexa-Fluor 488 anti-rabbit (dilution 1:200; Abcam) for 60 min at room temperature. Fluorescence intensity was visualized by microscopy (U-CMAD3; Olympus). In total, 125–150 fibres were measured in each sample. Capillary density was defined as total number of capillaries per CSA of the associated muscle fibres (Hepple et al., 2000; Hoppeler et al., 1981; Mathieu-Costello et al., 1988, 1991, 1992).

## Statistical analysis

Results are expressed at the mean ± SD. Normal distribution was verified with a Shapiro–Wilk test and a paired Student's *t* test was used to compare differences between two means. $P < 0.05$ was considered statistically significant. To assess within subject test–retest reliability, Pearson coefficient ($r$), the coefficient of variation (CV) and intraclass correlation coefficient (ICC) were calculated for $k$ measurements performed on different days. Correlation between respirometry variables and $k$ was performed to compare *ex vivo* and *in vivo* estimates of muscle oxidative capacity. Correlation between CD and $\Delta k$ was performed to examine the validity of NIRS to estimate $Dm_{O_2}$. Correlation analyses were expressed as Pearson coefficient ($r$). Prism, version 8.0 (GraphPad Software Inc., San Diego, CA, USA) was used for data analysis.

# Results

## Incremental exercise

$\dot{V}_{O_2peak}$ was 37.1 ± 8.0 mL min$^{-1}$ kg$^{-1}$ (range 22.0–50.2 mL min$^{-1}$ kg$^{-1}$) at 214 ± 52 W. Peak heart rate was 192 ± 9 beatsmin$^{-1}$, ~102% of the age-predicted maximum value. Respiratory exchange ratio was 1.28 ± 0.08; $[La]_b$ was 11.27 ± 2.24 mmol L$^{-1}$ and RPE was 19 ± 1. GET was 1.73 ± 0.41 L min$^{-1}$ (69% of $\dot{V}_{O_2peak}$), corresponding to 129 ± 34 W.

## Muscle oxygen uptake recovery rate constant by NIRS

In total, 48 m$\dot{V}_{O_2}$ recovery kinetics assessments were performed (24 in each of HIGH and LOW conditions) after 5 min of moderate intensity exercise (87 ± 26 W). Average $\dot{V}_{O_2}$ from last 60 s of exercise was 1.43 ± 0.38 L min$^{-1}$ and the respiratory exchange ratio was 0.95 ± 0.03. At the end of exercise, RPE was 10 ± 2 and $[La]_b$ was 1.47 ± 0.39 mmol L$^{-1}$. During last 20 s of cycling, quadriceps TSI averaged 62.7 ± 3.9% (corresponding to 50.4 ± 6.4% of PN).

$k_{HIGH}$ was 3.15 ± 0.45 min$^{-1}$ and 2.79 ± 0.55 min$^{-1}$ for the first and second repeat, respectively (range 1.88–4.01 min$^{-1}$). $k_{HIGH}$ was not different between

repeats ($P = 0.1057$). $k_{LOW}$ was 1.56 ± 0.79 min$^{-1}$ and 1.53 ± 0.57 min$^{-1}$ for the first and second repeat, respectively (range 0.38–2.65 min$^{-1}$). $k_{LOW}$ was not different between repeats ($P = 0.9299$). Coefficient of variation for repeated measurements was 26% and 12% for LOW and HIGH, respectively. In all participants and each repeat, $k_{HIGH}$ was greater than $k_{LOW}$ (both $P < 0.001$). The individual test–retest reliability of $k_{HIGH}$ and $k_{LOW}$, assessed on different days, was good ($r = 0.67$, $P < 0.001$; ICC = 0.68, CV = 19%) (Fig. 3). Having established reproducibility, the mean $k$ for each condition was calculated for comparison with biopsy variables. Mean ± SD $k_{HIGH}$ was 2.97 ± 0.36 min$^{-1}$ and mean $k_{LOW}$ was 1.54 ± 0.55 min$^{-1}$. $\Delta k$ ranged from 0.26 to 2.55 min$^{-1}$ with a mean ± SD value of 1.42 ± 0.69 min$^{-1}$. Immediately before the first cycling exercise, $\Delta[tot(Hb + Mb)]$ was 3.14 ± 0.77 μM and increased to 9.57 ± 4.52 μM and 11.20 ± 4.05 μM during arterial occlusions in HIGH and LOW respectively. During arterial occlusions, $\Delta[tot(Hb + Mb)]$ was significantly greater than rest ($P = 0.007$ and $P = 0.002$ for HIGH and LOW, respectively).

## Muscle mitochondrial respiration and capillarization

Measurements of mitochondrial O$_2$ flux in permeabilized muscle fibres are shown in Table 1. Maximal O$_2$ flux in phosphorylating state (CI + II$_P$) was 37.7 ± 10.6 pmol s$^{-1}$ mg$^{-1}$ (corresponding to 5.8 mL min$^{-1}$ 100 g$^{-1}$)

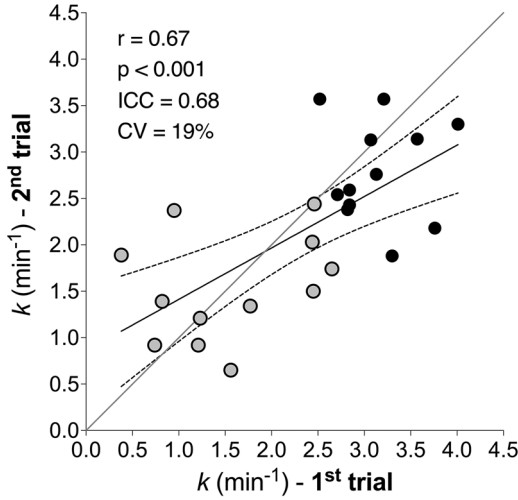

**Figure 3. Individual test–retest reliability of muscle oxidative capacity by NIRS**
The individual test–retest reliability of recovery rate constant ($k$) assessed by NIRS in different days during both HIGH (black circles) and LOW (grey circles) for 12 participants. Straight line represents linear regression curve and dashed line represent 95% confidence intervals. Grey line represents identity. Circles represent $k$ values obtained during first and second trial.

**Table 1. Mitochondrial $O_2$ flux in permeabilized muscle fibers**

| Mass specific (pmol $s^{-1}$ $mg^{-1}$) | $CI_L$ | $CI_P$ | $CI + II_P$ | $CI + II_E$ | $CII_E$ |
|---|---|---|---|---|---|
| Mean ± SD | 6.0 ± 3.2 | 14.6 ± 5.8 | 37.7 ± 10.6 | 56.8 ± 19.8 | 45.3 ± 14.8 |
| Minimum | 0.0 | 4.5 | 23.0 | 33.6 | 29.1 |
| Maximum | 11.3 | 23.9 | 54.3 | 99.9 | 76.6 |

Abbreviations: $CI_L$, leak respiration through CI; $CI_P$, maximum coupled mitochondrial respiration through CI; $CI + II_P$, maximum coupled mitochondrial respiration through CI + II; $CI + II_E$, maximum noncoupled mitochondrial respiration through CI + II; $CII_E$, maximum noncoupled mitochondrial respiration through CII. Mitochondrial respiration was measured in duplicate or triplicate ($n = 12$).

and maximal uncoupled $O_2$ flux ($CI + II_E$) was $56.8 \pm 19.8$ pmol $s^{-1}$ $mg^{-1}$ (corresponding to 8.8 mL $min^{-1}$ 100 $g^{-1}$). CS activity was $74.3 \pm 50.9$ mU $mg^{-1}$ protein.

Two typical images of CD capillary density measurement are shown in Fig. 4. Mean ± SD fibre CSA, fibre number and capillary number in each sample were $5083 \pm 73$ $\mu m^{-2}$, $134 \pm 47$ and $311 \pm 108$, respectively. CD was $469 \pm 73$ $mm^{-2}$, ranging from 348 to 586 $mm^{-2}$ among individuals.

### Correlations

$k_{HIGH}$ was correlated with $CI + II_P$ ($r = 0.80$, $P < 0.01$) and $CI + II_E$ ($r = 0.81$, $P < 0.01$). $k_{LOW}$ was not correlated with either $CI + II_P$ or $CI + II_E$ ($r = -0.10$ and $r = -0.11$, respectively) (Fig. 5A and B). $\Delta k$ was significantly correlated with capillary density ($r = -0.68$, $P = 0.015$) (Fig. 5C).

### Discussion

The present study tested the hypotheses that $m\dot{V}_{O_2}$ $k$ is associated with muscle oxidative capacity only in high TSI conditions and that the difference in $m\dot{V}_{O_2}$ $k$ between well oxygenated and poorly oxygenated muscle provides insight into $Dm_{O_2}$. Our data confirmed previous findings (Ryan et al., 2014) that, in well oxygenated tissue, $m\dot{V}_{O_2}$ recovery rate constant measured by NIRS,

$k_{HIGH}$, is correlated with maximal muscle $O_2$ flux in fibre bundles. Additionally, we showed for the first time that this relationship did not hold when tissue oxygenation was reduced below 50% of the physiological normalization, and $k_{LOW}$ was not associated with any variable describing muscle $O_2$ flux in fibre bundles. This finding is consistent with Fick's law in that $m\dot{V}_{O_2}$ becomes increasingly dependent on $Dm_{O_2}$ as the microvascular-to-myocyte $P_{O_2}$ difference is reduced (see eqn (1)). From this, we demonstrated that $\Delta k$, the difference between $k_{HIGH}$ and $k_{LOW}$, was associated with capillary density in biopsy samples of the same tissue, a primary structural determinant of $Dm_{O_2}$. Thus, the NIRS-derived response measured under conditions of both well oxygenated and poorly oxygenated skeletal muscle provided a non-invasive means of assessing both muscle oxidative capacity and muscle diffusing capacity *in vivo*. The application of this NIRS protocol may be of great interest for the investigation of skeletal muscle oxidative and diffusing capacities in response to exercise training or in disease states (e.g. heart failure, chronic obstructive pulmonary disease, myopathies).

### NIRS and muscle oxidative capacity

In health, skeletal muscle oxidative capacity is strongly correlated with whole body aerobic capacity and exercise

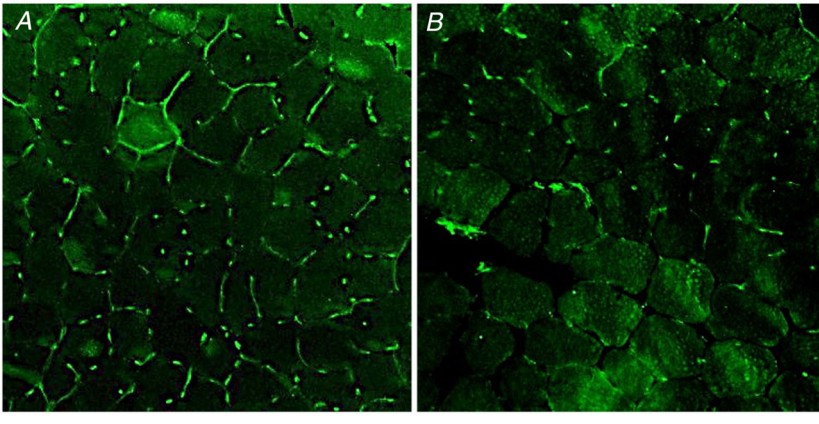

**Figure 4. Muscle capillarization from two participants**
Immunofluorescence identification of capillaries in cross-sections of *vastus lateralis* from two participants. *A*, participant with the highest capillary density. *B*, participant with the lowest capillary density. After immunostaining using anti-CD31, capillaries appear green ($n = 2$).

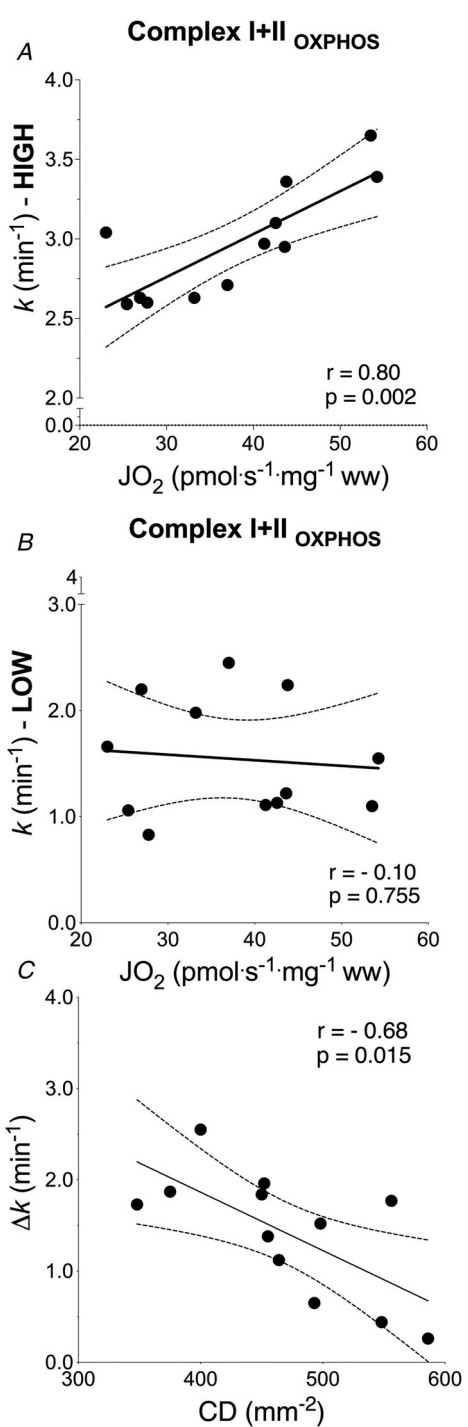

**Figure 5. Association between *in vivo* and *ex vivo* estimation of skeletal muscle OXPHOS capacity and capillary density**
Association between *in vivo* (NIRS) and *ex vivo* (biopsy) estimation of (*A* and *B*) skeletal muscle OXPHOS capacity and (*C*) skeletal muscle capillary density. Correlation between muscle oxidative phosphorylation capacity by high-resolution respirometry and recovery rate constant by NIRS measured in HIGH (*A*) and LOW (*B*) conditions. *C*, correlation between $\Delta k$ ( $= k_{HIGH} - k_{LOW}$) by NIRS and capillary density (CD) from biopsy. $\Delta k$ is lower in participants with a greater muscle capillarization. Linear regression (solid) 95% confidence intervals (dash) (*n* = 12).

performance (Holloszy, 1967; Hood et al., 2011; Hoppeler et al., 1985). Moreover, muscle oxidative capacity and mitochondrial function are impaired in conditions of physical inactivity (Buso et al., 2019; Zuccarelli et al., 2021), aging (Layec et al., 2013) and chronic disease, such as obesity (Lazzer et al., 2013; Menshikova et al., 2005), diabetes (Joseph et al., 2012) myopathy (Grassi et al., 2019, 2020), pulmonary obstructive disease (Adami et al., 2017, 2020) and neuromuscular disease (Breuer et al., 2013). Traditionally, muscle oxidative capacity has been studied using *ex vivo* approaches involving muscle biopsy samples and measurement of enzyme activity or mitochondrial respiratory capacity in isolated mitochondrial preparations and permeabilized muscle fibres (Brand & Nicholls, 2011; Chance & Williams, 1955; Gnaiger, 2009; Holloszy, 1967; Perry et al., 2013). For a long time, *in vivo* approaches were limited to [31]P nuclear magnetic resonance spectroscopy for measuring the recovery rate of phosphocreatine after exercise, which is associated with maximal $O_2$ flux in muscle (Blei et al., 1993; Kemp et al., 1993; Kent & Fitzgerald, 2016). Over the past decade, however, NIRS coupled with intermittent arterial occlusions, was shown to be a valuable tool for assessing the $m\dot{V}_{O_2}$ recovery rate constant, $k$, which is directly associated with muscle oxidative capacity (Adami & Rossiter, 2018; Adami et al., 2017; Grassi & Quaresima, 2016; Hamaoka & McCully, 2019; Hamaoka et al., 1996; Motobe et al., 2004; Ryan et al., 2012). The estimation of muscle oxidative capacity using NIRS has been already examined in upper and lower limb muscles and in both healthy subjects and patients affected by chronic diseases (Adami et al., 2017; Erickson et al., 2013, 2015; Harp et al., 2016; Meyer, 1988; Ryan et al., 2013, 2014; Willingham & McCully, 2017). However, muscle oxidative capacity estimation by NIRS relies on two main assumptions (Adami & Rossiter, 2018; Chung et al., 2018): (i) muscle contractions are sufficient to maximally activate mitochondrial oxidative enzymes and (ii) $O_2$ concentration at skeletal muscle level is not a limiting factor to oxidative phosphorylation. Regarding the first point, it has been demonstrated that the first-order relationship between phosphocreatine dynamics and ATP production by oxidative phosphorylation is valid only when mitochondrial oxidative enzymes are maximally activated. Additionally, experiments on isolated single frog muscle fibres show that maximal activation of mitochondrial enzymes may not be achieved when muscle stimulation (or contraction frequency) is too low (Wüst et al., 2013). For this reason, we used 5 min of moderate intensity cycling to stimulate OXPHOS and activate a wide range of regulated enzymes within the mitochondrial matrix, with the aim of reaching a high, ideally, maximal mitochondrial activation. Our data for $k_{HIGH}$ show a good correlation with maximal $O_2$ flux in biopsy samples (either phosphorylating or uncoupled), consistent with previous

findings (Ryan et al., 2014). These data confirm that NIRS provides a reasonable non-invasive estimate of muscle oxidative capacity when the tissue is well oxygenated.

The second assumption, that $O_2$ concentration is not limiting in the NIRS test, has received relatively less attention. Because $m\dot{V}_{O_2}$ depends in part on the $O_2$ pressure difference between the microvasculature and the inner mitochondrial membrane in the myocyte to facilitate $O_2$ diffusion, it stands to reason that reducing $Pmv_{O_2}$ could reduce $O_2$ flux and limit $m\dot{V}_{O_2}$ $k$. This suggestion was confirmed previously by the reduction in $m\dot{V}_{O_2max}$ and phosphocreatine recovery rate constant under conditions of reduced $Pmv_{O_2}$ imposed by breathing hypoxic gas mixtures (Haseler et al., 1999, 2004; Richardson et al., 1999). By reducing ($Pmv_{O_2}$ – $Pim_{O_2}$) in hypoxia, $m\dot{V}_{O_2}$ becomes increasingly dependent on $Dm_{O_2}$ (eqn (1)). Despite this, no study to date has investigated whether NIRS-derived $m\dot{V}_{O_2}$ $k$ estimation is affected by changes in $O_2$ availability. We investigated the reliability and validity of NIRS-derived $m\dot{V}_{O_2}$ $k$ to reflect muscle oxidative capacity under conditions of well and poorly oxygenated muscle. As expected, $k_{HIGH}$ was associated with maximal $O_2$ flux in fibre bundles performed in a hyperoxic environment (i.e. in non-limiting $O_2$ availability), when TSI was maintained above 50% of PN ($r = 0.80–0.81$) (Fig. 5A) and showed good reproducibility (test–retest $r = 0.67$, ICC = 0.) (Fig. 3). The regression coefficient between methods was very similar to that observed by Ryan et al. (2014) (Ryan et al., 2014). However, our data also show that this association was lost ($r = -0.11$ to $-0.10$) when the arterial occlusion protocol was experimentally manipulated to hold TSI between 0% and 10% PN. In the $k_{LOW}$ condition, good test–retest reproducibility was maintained, but the association with maximal $O_2$ flux by tissue respirometry was absent (Fig. 5B).

The two NIRS tests assessed the same muscle region in tests applied a few minutes apart, therefore differences between $k_{HIGH}$ and $k_{LOW}$ could not reasonably reflect structural or functional properties in the skeletal muscle mitochondria. Rather, the difference reflects the increasing importance of $Dm_{O_2}$ in $k_{LOW}$, when ($Pmv_{O_2}$ – $Pim_{O_2}$) was reduced by experimental manipulation of the occlusion protocol. Thus, $m\dot{V}_{O_2}$ $k$ is only a valid method to estimate muscle oxidative capacity when assessed as $k_{HIGH}$ (i.e. above 50% PN) (Adami & Rossiter, 2018).

### NIRS and muscle $O_2$ diffusion

As previously described, $m\dot{V}_{O_2}$ is dependent on the interplay between both convective and diffusive transport (Roca et al., 1992). In our protocol, we estimated muscle oxidative capacity from the TSI slope during repeated arterial occlusions, which minimizes the effects of $O_2$ delivery on the estimation of muscle oxidative capacity. Moreover, we effectively manipulated arterial occlusions such that the sum of ($Pmv_{O_2}$ – $Pim_{O_2}$) was either large or small, in the HIGH and LOW conditions, respectively. In these conditions, convective $O_2$ transport may have an influence on our surrogate measure of $Dm_{O_2}$ by modifying capillary Hb volume, RBC transit time or RBCs immediately adjacent to active muscle fibres during the reperfusion phase between two occlusions. However, the absence of differences in $\Delta$[tot(Hb + Mb)] between HIGH and LOW suggests that, under these strictly controlled experimental conditions, convective $O_2$ transport did not influence $m\dot{V}_{O_2}$ $k$.

By measuring $m\dot{V}_{O_2}$ under these two conditions, we could simultaneously solve by elimination for $Dm_{O_2}$ (eqn (1)). Using NIRS the absolute value of ($Pmv_{O_2}$ – $Pim_{O_2}$) is unknown; rather the ranges of TSI used reflected two relative oxygenation values (i.e. HIGH and LOW). Also, absolute values of $m\dot{V}_{O_2}$ are not precisely known using NIRS, although they may be estimated assuming values for tissue [Hb + Mb] (among other assumptions). Nevertheless, the NIRS-based protocol we used allows for relative measurement of $m\dot{V}_{O_2}$ and ($Pmv_{O_2}$ – $Pim_{O_2}$), such that solving eqn (1) for a relative estimate of $Dm_{O_2}$ is possible. We used the $k$ value for $m\dot{V}_{O_2}$, rather than $m\dot{V}_{O_2}$ itself, because, as demonstrated, $k_{HIGH}$ reflects muscle oxidative capacity and because the kinetics of $m\dot{V}_{O_2}$ are unrelated to absolute measurements (thereby reducing potential variability introduced by comparing absolute $m\dot{V}_{O_2}$ values from NIRS). Assessment of $Dm_{O_2}$ was performed by capillary density and the finding that $\Delta k$ was correlated with CD (Fig. 5C) supports the validity of NIRS-based protocol to estimate $Dm_{O_2}$.

Although CD does not assess length and diameter of vessels, the measurement of capillary density in a muscle cross-section may provide information to understand the $O_2$ diffusing capacity of skeletal muscle (Hepple et al., 2000; Poole et al., 2020, 2021, 2022; Saltin & Gollnick, 1983). Our CD values ranged from 348 to 586 mm$^{-2}$, in line with published data (Poole et al., 2020). On the other hand, $\Delta k$ ranged more widely from 0.26 to 2.55 min$^{-1}$. This is presumably because $\Delta k$ is also influenced by muscle oxidative capacity; therefore, changes in $\Delta k$ under fixed TSI conditions may vary more widely than capillary structure alone. Nevertheless, the finding of a significant association between $\Delta k$ and CD may allow a valid non-invasive estimation of CD *in vivo*.

The ability to measure $O_2$ dynamics in microvessels in human skeletal muscles is currently beyond reach (Koga et al., 2014; Lundby & Montero, 2015). Previous studies have attempted this by measuring structural components related to the path through plasma and capillary wall to the cytoplasm (e.g. capillary density) or calculating the ratio of $m\dot{V}_{O_2}$ and $O_2$ pressure gradient between microvessels and mitochondria using methods developed previously

(Hepple et al., 2000; Richardson et al., 1999; Roca et al., 1989, 1992). Although these elegant experiments were fundamental to move forward knowledge in the field, they do not fully account for the physiological mechanisms of *in vivo* regulation and reduce a complex diffusion process to a few key variables (Hepple et al., 2000).

A step towards a more physiological approach was used by Brown et al. (2001). In endurance and resistance trained athletes, capillary filtration capacity was estimated with plethysmography during small incremental steps in venous occlusion pressure. An association was found between filtration capacity and muscle capillarity, consistent with a relationship between capillary filtration and capillary surface area (Brown et al., 2001; Hunt et al., 2013). In the present study, we used NIRS to follow the flux of $O_2$ rather than fluid within the capillary-to-myofiber. In addition, NIRS is able to estimate changes in $\Delta[\text{tot}(\text{Hb} + \text{Mb})]$ within the microvasculature, such that our NIRS-based method probably more closely reflects apposition of red blood cells with capillary endothelium and therefore the $O_2$ diffusion pathway. The action of repeated occlusions has the effect of increasing $\Delta[\text{tot}(\text{Hb} + \text{Mb})]$ in the microvasculature ($\sim$10 μm in $\Delta[\text{tot}(\text{Hb} + \text{Mb})]$ from rest which should account for an increase in haematocrit of $\sim$18%, leading to a total haematocrit of 33%), thereby generating a diffusional surface area that may be closer to *in vivo* condition when $Dm_{O_2}$ is maximized (i.e. high longitudinal capillary recruitment at approach exercise at maximal aerobic capacity). That NIRS measures the oxygenation of all [Hb + Mb] within the field of view may therefore be a more valid reflection of the final steps in the $O_2$ cascade than structural properties of the capillaries alone. However, because this remains undeveloped, more advanced techniques are needed to test this hypothesis.

The correlation coefficients of $k$ with muscle oxidative capacity and $\Delta k$ with CD were relatively small compared to correlations for clinical techniques (Hanneman, 2008). However, the Pearson correlation coefficient in the present study ($r = 0.80$ and $-0.68$, respectively) was similar with other validation studies comparing *in vivo* methods with *ex vivo* approaches using NIRS ($r = 0.61$–0.74) (Ryan et al., 2014). Although NIRS and biopsy were taken from the same muscle region, the volumes assayed by each of these techniques are different ($\sim$2–3 $cm^3$ and $\sim$1 $mm^3$, respectively). In addition, the structural and functional assessments differ between NIRS and biopsy. Unlike NIRS, maximal $O_2$ flux in fibre bundles in respirometry is assessed using supraphysiological concentrations of substrates, including $O_2$, and reflects all fibres in the sample. Maximal $O_2$ flux based on NIRS signals derive from only those fibres that were active during the exercise, and therefore may be biased towards low order fibres (slow, oxidative). As discussed, CD is a structural contributor to $Dm_{O_2}$, but may not reflect to the true capacity for

$O_2$ diffusion in tissue with variable capillary content of red blood cells. Therefore, variability between methods is expected, with the NIRS potentially having the advantage that it assays a larger muscle volume than the biopsy, and also assays under physiological conditions *in vivo*.

## Conclusions

In summary, we report the reliability and accuracy of NIRS measurements of $m\dot{V}_{O_2}$ during intermittent arterial occlusions protocols for estimating muscle oxidative capacity and muscle $Dm_{O_2}$ *in vivo*. The $m\dot{V}_{O_2}$ recovery rate constant, $k$, was a reasonable and reliable method to assess muscle oxidative capacity only when assessed in well oxygenated quadriceps muscle ($>$50% of TSI physiological normalization). Experimental manipulation of TSI through timing and duration of intermittent arterial occlusions provided a new variable, $\Delta k$, which was the difference in $k$ between well oxygenated and poorly oxygenated experimental conditions. As hypothesized, we found that $\Delta k$ was associated with CD, a structural determinant of $Dm_{O_2}$. Therefore, the NIRS-based protocol we describe represents a cost-effective and non-invasive means of assessing both muscle oxidative capacity and muscle $O_2$ diffusive capacity *in vivo*.

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

## Additional information

### Data availability statement

All relevant data are presented as individual data points in the figures. Data not presented with individual data points are available from the corresponding author upon reasonable request.

### Competing interests

The authors declare that they have no competing interests.

### Author contributions

AP, AA, RM, LB, EC, RB, BG, HR and SP were responsible for the conception or design of the work. AP, AA, RM, LB, EC, LZ, MAP, HR and SP were responsible for acquisition or analysis or interpretation of data for the work. AP, AA, RM, LB, EC, LZ, MAP, RB, BG, HR and SP were responsible for drafting the work or revising it critically for important intellectual content. AP, AA, RM, LB, EC, LZ, MAP, RB, BG, HR and SP were responsible for final approval of the version submitted for publiction. AP, AA, RM, LB, EC, LZ, MAP, RB, BG, HR and SP agree to be accountable for all aspects of the work.

### Funding

Simone Porcelli was supported by a grant from Sports Medicine Italian Federation (FMSI01092021). Alessandra

Adami is supported by a grant from NIH (R01HL151452). Harry Rossiter is supported by grants from NIH (R01HL151452, R01HL153460, P50HD098593, R01DK122767, P2CHD086851), the Tobacco Related Disease Research Program (T31IP1666) and the University of California, Office of the President. He reports consulting fees from Omniox Inc., and is involved in contracted clinical research with Boehringer Ingelheim, GlaxoSmithKline, Novartis, AstraZeneca, Astellas, United Therapeutics, Genentech and Regeneron. He is a visiting Professor at the University of Leeds, UK.

## Acknowledgements

We thank the participants who joined the project, as well as graduate students Andrea Alberti and Andrea Colombo for their help during data recording.

Open Access Funding provided by Universita degli Studi di Pavia within the CRUI-CARE Agreement.

## Keywords

biopsy, capillary density, mitochondria, recovery kinetics, skeletal muscle

## Supporting information

Additional supporting information can be found online in the Supporting Information section at the end of the HTML view of the article. Supporting information files available:

**Statistical Summary Document**
**Peer Review History**

