## [Peer Review History · The Journal of Physiology]

Near-infrared spectroscopy estimation of combined skeletal muscle oxidative capacity and O₂ diffusion capacity in humans

Andrea Pilotto, Alessandra Adami, Raffaele Mazzolari, Lorenza Brocca, Emanuela Crea, Lucrezia Zuccarelli, Maria Antonietta Pellegrino, Roberto Bottinelli, Bruno Grassi, Harry B Rossiter, and Simone Porcelli
DOI: 10.1113/JP283267

Corresponding author(s): Simone Porcelli (simone.porcelli@unipv.it)

The following individual(s) involved in review of this submission have agreed to reveal their identity: David C Poole (Referee #1); Takafumi Hamaoka (Referee #2)

Review Timeline:

Submission Date:	28-Apr-2022
Editorial Decision:	24-May-2022
Revision Received:	04-Jul-2022
Editorial Decision:	14-Jul-2022
Revision Received:	15-Jul-2022
Accepted:	27-Jul-2022

Senior Editor: Michael Hogan

Reviewing Editor: Ross Pollock

Transaction Report:

Dear Dr Porcelli,

Re: JP-RP-2022-283267 "Near-infrared spectroscopy estimation of combined skeletal muscle oxidative capacity and O₂ diffusion capacity in humans" by Andrea Pilotto, Alessandra Adami, Raffaele Mazzolari, Lorenza Brocca, Emanuela Crea, Lucrezia Zuccarelli, Maria Antonietta Pellegrino, Roberto Bottinelli, Bruno Grassi, Harry B Rossiter, and Simone Porcelli

Thank you for submitting your manuscript to The Journal of Physiology. It has been assessed by a Reviewing Editor and by 2 expert Referees and I am pleased to tell you that it is considered to be acceptable for publication following satisfactory revision.

The reports are copied at the end of this email. Please address all of the points and incorporate all requested revisions, or explain in your Response to Referees why a change has not been made.

NEW POLICY: In order to improve the transparency of its peer review process The Journal of Physiology publishes online as supporting information the peer review history of all articles accepted for publication. Readers will have access to decision letters, including all Editors' comments and referee reports, for each version of the manuscript and any author responses to peer review comments. Referees can decide whether or not they wish to be named on the peer review history document.

Authors are asked to use The Journal's premium BioRender (<https://biorender.com/>) account to create/redraw their Abstract Figures. Information on how to access The Journal's premium BioRender account is here:

<https://physoc.onlinelibrary.wiley.com/journal/14697793/biorender-access> and authors are expected to use this service. This will enable Authors to download high-resolution versions of their figures. The link provided should only be used for the purposes of this submission. Authors will be charged for figures created on this premium BioRender account if they are not related to this manuscript submission.

I hope you will find the comments helpful and have no difficulty returning your revisions within 4 weeks.

Your revised manuscript should be submitted online using the links in Author Tasks Link Not Available.

Any image files uploaded with the previous version are retained on the system. Please ensure you replace or remove all files that have been revised.

REVISION CHECKLIST:

- Article file, including any tables and figure legends, must be in an editable format (eg Word)
- Abstract figure file (see above)
- Statistical Summary Document
- Upload each figure as a separate high quality file
- Upload a full Response to Referees, including a response to any Senior and Reviewing Editor Comments;
- Upload a copy of the manuscript with the changes highlighted.

- A potential 'Cover Art' file for consideration as the Issue's cover image;
- Appropriate Supporting Information (Video, audio or data set https://jp.msubmit.net/cgi-bin/main.plex?form_type=display_requirements#supp).

To create your 'Response to Referees' copy all the reports, including any comments from the Senior and Reviewing Editors, into a Word, or similar, file and respond to each point in colour or CAPITALS and upload this when you submit your revision.

I look forward to receiving your revised submission.

If you have any queries please reply to this email and staff will be happy to assist.

Yours sincerely,

Michael C. Hogan
Senior Editor
The Journal of Physiology
<https://jp.msubmit.net>
<http://jp.physoc.org>
The Physiological Society
Hodgkin Huxley House
30 Farringdon Lane
London, EC1R 3AW
UK
<http://www.physoc.org>
<http://journals.physoc.org>

REQUIRED ITEMS:

- Author photo and profile. First (or joint first) authors are asked to provide a short biography (no more than 100 words for one author or 150 words in total for joint first authors) and a portrait photograph. These should be uploaded and clearly labelled with the revised version of the manuscript. See Information for Authors for further details.
- The contact information provided for the person responsible for 'Research Governance' at your institution is an author on this paper. Please provide an alternative contact who is not an author on this paper or confirm that the author whose email was provided has sole responsibility for research governance. This is the person who is responsible for regulations, principles and standards of good practice in research carried out at the institution, for instance the ethical treatment of animals, the keeping of proper experimental records or the reporting of results.
- The Journal of Physiology funds authors of provisionally accepted papers to use the premium BioRender site to create high resolution schematic figures. Follow this link and enter your details and the manuscript number to create and download figures. Upload these as the figure files for your revised submission. If you choose not to take up this offer we require figures to be of similar quality and resolution. If you are opting out of this service to authors, state this in the Comments section on the Detailed Information page of the submission form. The link provided should only be used for the purposes of this submission. Authors will be charged for figures created on this premium BioRender account if they are not related to this manuscript submission.
- Please upload separate high-quality figure files via the submission form.
- Please ensure that any tables are in Word format and are, wherever possible, embedded in the article file itself.
- A Statistical Summary Document, summarising the statistics presented in the manuscript, is required upon revision. It must be on the Journal's template, which can be downloaded from the link in the Statistical Summary Document section here: https://jp.msubmit.net/cgi-bin/main.plex?form_type=display_requirements#statistics
- Please include an Abstract Figure. The Abstract Figure is a piece of artwork designed to give readers an immediate understanding of the research and should summarise the main conclusions. If possible, the image should be easily 'readable' from left to right or top to bottom. It should show the physiological relevance of the manuscript so readers can assess the importance and content of its findings. Abstract Figures should not merely recapitulate other figures in the manuscript. Please try to keep the diagram as simple as possible and without superfluous information that may distract from the main conclusion(s). Abstract Figures must be provided by authors no later than the revised manuscript stage and should be uploaded as a separate file during online submission labelled as File Type 'Abstract Figure'. Please ensure that you include the figure legend in the main article file. All Abstract Figures should be created using BioRender. Authors should use The Journal's premium BioRender account to export high-resolution images. Details on how to use and access the premium account are included as part of this email.

EDITOR COMMENTS

Reviewing Editor:

The authors have presented a very interesting manuscript which both reviewers have highlighted the novelty of the research and the potential impact it may have in the field. Overall the authors have developed a methodology which can provide an alternative means to assess muscle oxidative capacity and oxygen diffusion capacity. Some concerns with the manuscript have been raised by the reviewers particularly around certain aspects of the analysis, in particular the use of C:F to estimate DmO₂, which should be addressed.

Line 314, 284 - Please ensure where appropriate all P values are given precisely except for when <0.001.

REFEREE COMMENTS

Referee #1:

GENERAL COMMENTS

This paper, by an extremely accomplished covey of investigators, tested the novel hypotheses that muscle VO₂ recovery kinetics after exercise under well and poorly oxygenated conditions could provide unique insights into muscle O₂ diffusing capacity and its determinants. The data presented have a high degree of novelty and some of the concepts explored, and the manner of that exploration, are on the cutting edge of scientific investigation in this field. That said there are concerns regarding the methodology, analyses and interpretation, as well as the literature cited, attention to which would substantially improve the validity, interpretation and putative impact of this most interesting paper.

SPECIFIC COMMENTS

LINE(s)

19 what about the interplay between convective and diffusive properties as both are impacted by the ischemic condition imposed (see Roca et al. (J Appl Physiol (1985). 1992 Sep;73(3):1067-76) for exemplar of interdependence)?

21 Here and subsequently I believe substantial error in the estimate of DmO₂ may have arisen from use of C:F which assesses well changes of capillarity across interventions such as training or disuse, but weakens the analyses when used herein. Briefly, as considered by Federspiel and Popel (Microvasc Res. 1986 Sep;32(2):164-89) in their elegant modeling approach and substantiated by a range of physiological assessments subsequently, the primary determinant of Dm is the number of RBCs immediately adjacent to the contracting muscle fibers (in RBC flowing capillaries) at any given instant. In individuals with no difference in muscle fiber cross-sectional area (XSA) C:F this provides an index of capillarity differences that might reflect Dm as considered by F&P. However, when individuals vary in fiber XSA markedly as appears to be the case for the subjects pictured in Figure 4 herein it does not. It is recommended that the authors consult a morphometrist with experience in capillary measurements and make high-quality measurements of capillary density and used these to estimate capillary volume density (using accepted measurements of capillary diameter, tortuosity etc. see Mathieu-Costello and colleagues beautiful work on capillary geometry and such from the 1980s-1990s (e.g., Microvasc Res. 1988 Jul;36(1):40-55; Am J Physiol. 1991 Nov;261(5 Pt 2):H1617-25). This will help negate the confounding influence of fiber XSA from estimates of Dm within and among subjects and better reflect the existent Dm.

25 To help the reader reconcile the present measurements with in vivo and other studies, please include measurement of O₂ flux in ml/min/100 g or such - possibly in addition to that presented now (here and throughout).

28 Why was citrate synthase activity (or such) not measured on the biopsies?

52 Probably cite Richardson et al. 1995 here as well as Clanton.

54 Whether capillary-to-mitochondrial diffusion distances in skeletal muscle are of any importance at all has been seriously questioned by the data from Honig and colleagues (e.g., Adv Exp Med Biol. 1986;200:487-94), Hepple et al. (J Appl Physiol (1985). 2000 Feb;88(2):560-6) as well as Richardson and colleagues' proton-MRS work and the development of novel models of capillary function that incorporate the latest findings in this rapidly-developing field (Poole et al. Eur J Appl Physiol. 2022 Jan;122(1):7-28).

58 Please consider adding Roca et al. (1992) in with Richardson et al. here as above.

69 For ease of reading, please state 2 unknowns here.

87 Impressive though S&G, 1983 is, subsequent work on capillary: fiber surface ratio and especially capillary volume density - as it impacts both capillary RBC transit time and the instantaneous RBC number and Hb volume in approximation with the muscle fibers - represent probably a far more important index of DmO₂ (see above).

104 Given the minimal extra work required and the Cores of Reproducibility initiatives on this precise topic it was disappointing that the authors did not validate a VO₂max (J Appl Physiol (1985). 2017 Apr 1;122(4):997-1002). Maximal HR and RER are not adequate indicators of such.

120 Perhaps I'm being wooden here but, at this location and subsequently, the "0-10% PN does not appear to cohere with either Figure 1 (LOW is 40-50% TSI as indicated) or subsequent data and text. Please clarify and make internally consistent.

144 Clarify when (i.e., at rest) that the initial value was set to "0" and, perhaps include oxy and deoxy Mb in parentheses. As

I suspect Professor Shunsaku Koga would agree TRS would have been better than CWSR-NIRS.

147 Define "SRS" if not already.

157 "tHb+tMb"

165 As above, not clear what or where 0-10% of PN is.

167 Much appreciated that a test/retest was performed.

Figure 2 - perhaps simplify the time axis - or does 1550 s have some other significance? Identify PN point and also consider switching panels C and D so they line up below the appropriate TSI graphs.

186 Only 1 grey point I could see.

218 Please present chamber PO₂ as well as concentration to allow reader to appreciate how these may relate to physiologic values (or not!).

234 As above. Please also put these in ml O₂/min/100 g or kg.

249 More details of morphometry needed as well as measurement of fibers XSA, capillary density and estimate of capillary volume density.

272 Unclear. Was PN then ~125%? Please clarify as requested above.

275 The lack of significant difference of mean values between two tests just might be evidence for no overarching effect of a "practice" run. Please present the coefficient of variation to assess better 'reproducibility.'

279 A correlation of 0.67 really only explains 45% of the variability. What factors might contribute to the "other 55%"?

282 "tHb+tMb" was 3.14 micromole - wasn't this set to "0"? please explain.

284 How much Hb volume does the ~6 micromoles of elevated [Hb] account for? This is a critical calculation because, combined with the suggested estimation of capillary volume density, it allows the mean impact of exercise and occlusion on capillary hematocrit - and therefore DmO₂ - to be estimated. For example, supposing that the capillary density is 300 mm⁻¹ (allowing for capillary tortuosity and branching and a mean capillary diameter of 6 microns, capillary volume density would be 1.1%) and Hcrit were presumed to be ~30%. It can be calculated that the 6 micromole (0.39 g Hb per L) increase in [Hb] would elevate capillary Hcrit from 30-34% which is, I think, quite reasonable, and adds a practical and logical quantitative coherence to the data interpretation. Perhaps enlisting a muscle morphometry specialist to help with this?

Figure 3. Please state c.v.

Figure 5. What is the correlation between K HIGH and C:F (or better - capillary volume density).

327 and 335 As above....better for cap vol density.

330 The implications of this work, as several of the authors have stated in previous investigation, extends crucially to heart failure. That point could be made usefully here.

343 "function" is meant surely?

394 Perhaps discuss briefly the possible concerns with hyperoxic respirometry?

413 "agnostic" is there a better word/phrase?

415-6 Seems circular logic.

418-25 The heart presents a good exemplar of some of the problems, discussed above, with using C:F ratio as an index of capillarity (and therefore estimate of Dm). Myocardial C:F is ~1.1 and yet, because of the very small myocardial fiber XSA myocardial capillary density and volume density, and therefore, Dm are extremely high.

433 This seems a bit harsh on the great Peter Wagner whose relevant experiments were, inarguably, groundbreaking and moved this field forward considerably.

442

445-7 As above provide quantitative reasoning.

449 "more advanced" might be qualified to "as yet undeveloped".

456 Please state relative volumes for biopsy vs NIRS.

464 What effect does only recruiting a modest portion of the muscle interrogated have on the measured kinetics/absolute levels measured?

469 Perhaps "validity" assessment should be left to the reader: "reasonable" would be better here?

Referee #2:

General Comments

This study examined if the recovery rate constant of mVO_2 in high tissue saturation index (TSI) is associated with muscle oxidative capacity and if the difference in the recovery rate constant of mVO_2 between non- O_2 limiting (HIGH) and O_2 limiting (LOW) TSI conditions. This study has been well-designed and elaborately conducted and contains invaluable outcomes. However, some of the theoretical basis should be thoroughly explained for readers to understand.

Major Comments

The authors thought that they would be able to solve for diffusion capacity of O_2 (DmO_2) by simultaneous subtraction of the two unknowns in Eq. 1. It is true that intramyocyte PO_2 ($P_{im}O_2$) should be almost constant during moderate- to high-intensity of exercise. However it is evident that microvascular PO_2 ($P_{mv}O_2$) in HIGH and LOW conditions in this study is not equivalent as shown in Figure 2 (although TSI is not a direct measure of PO_2). The authors should explain this issue, for example, by showing an equation of this estimation.

The authors should discuss the NIR signal contribution from Hb and Mb and the influence of the Mb signal contribution on the assumption used in this study.

It seems that the authors did not mention a randomization of LOW and HIGH conditions. Are there any "order effect" on LOW and HIGH conditions. It is probable that the 1st trial of exercise would facilitate the oxidative capacity in the 2nd trial.

As the correlation between JO_2 and k -LOW is insignificant and rather "flat", the significant correlation between C:F and Δk is primarily derived from the correlation of JO_2 to k -HIGH. How about the correlation between C:F and k -HIGH? It is not surprising to find a good correlation between C:F and k -HIGH because VO_{2max} and capillarity correlates well.

The authors should discuss a significance of the O_2 level in LOW (0-10%) and HIGH (50-60%). Why this level was used.

In figure 3, it is suggested that LOW and HIGH plots be distinct by using a different plot design such as filled and unfilled circles.

Minor Comments

L 112: VISIT 3 should be the 2nd TRIAL.

L 274: need (,) before respectively.

END OF COMMENTS

Confidential Review

28-Apr-2022

EDITOR COMMENTS

Reviewing Editor:

The authors have presented a very interesting manuscript which both reviewers have highlighted the novelty of the research and the potential impact it may have in the field. Overall the authors have developed a methodology which can provide an alternative means to assess muscle oxidative capacity and oxygen diffusion capacity. Some concerns with the manuscript have been raised by the reviewers particularly around certain aspects of the analysis, in particular the use of C:F to estimate DmO₂, which should be addressed.

Dear Editor, thanks for giving us the opportunity to revise the manuscript and we thank the reviewers for their comments. We are happy to know that our study was appreciated, and we revised the manuscript according to reviewers' suggestions. Some new analyses were also performed to improve ex-vivo parameters used to estimate DmO₂.

A detailed list of responses to the reviewers' suggestions follows. Line numbers refer to the red lined version of the manuscript.

Line 314, 284 - Please ensure where appropriate all P values are given precisely except for when <0.001.

Thank you. The appropriate p values are now included as suggested at lines 338 and 374.

Referee #1:

GENERAL COMMENTS

This paper, by an extremely accomplished covey of investigators, tested the novel hypotheses that muscle VO₂ recovery kinetics after exercise under well and poorly oxygenated conditions could provide unique insights into muscle O₂ diffusing capacity and its determinants. The data presented have a high degree of novelty and some of the concepts explored, and the manner of that exploration, are on the cutting edge of scientific investigation in this field. That said there are concerns regarding the methodology, analyses and interpretation, as well as the literature cited, attention to which would substantially improve the validity, interpretation and putative impact of this most interesting paper.

We thank the reviewer for the time dedicated in reviewing our manuscript. We are happy to know that our study raised your interest. Comments were helpful and provided the opportunity to deepen our knowledge in the field of capillarization and muscle diffusion.

Seminal works of Dr. Mathieu-Costello and Prof. Hoppeler were used, as suggested, to address the issues raised and perform new analyses to improve DmO₂ estimation from our samples by calculating capillary density. We also provided further details, where possible. We believe the revision improved greatly the quality of the manuscript.

SPECIFIC COMMENTS

LINE(s)

19 what about the interplay between convective and diffusive properties as both are impacted by the ischemic condition imposed (see Roca et al. (J Appl Physiol (1985). 1992 Sep;73(3):1067-76) for exemplar of interdependence)?

We agree with the reviewer that muscle oxygen uptake is influenced by the interplay between both convective and diffusive components. We did not change the text in the abstract due to the word limit. However, we added a paragraph in the introduction (Lines 47-48) and in the discussion (Lines 467-476) to better clarify this aspect.

21 Here and subsequently I believe substantial error in the estimate of DmO₂ may have arisen from use of C:F which assesses well changes of capillarity across interventions such as training or disuse, but weakens the analyses when used herein. Briefly, as considered by Federspiel and Popel (Microvasc Res. 1986 Sep;32(2):164-89) in their elegant modeling approach and substantiated by a range of physiological assessments subsequently, the primary determinant of Dm is the number of RBCs immediately adjacent to the contracting muscle fibers (in RBC flowing capillaries) at any given instant. In individuals with no difference in muscle fiber cross-sectional area (XSA) C:F this provides an index of capillarity differences that might reflect Dm as considered by F&P. However, when individuals vary in fiber XSA markedly as appears to be the case for the subjects pictured in Figure 4 herein it does not. It is recommended that the authors consult a morphometrist with experience in capillary measurements and make high-quality measurements of capillary density and used these to estimate capillary volume density (using accepted measurements of capillary diameter, tortuosity etc. see Mathieu-Costello and colleagues beautiful work on capillary geometry and such from the 1980s-1990s (e.g., Microvasc Res. 1988 Jul;36(1):40-55; Am J Physiol. 1991 Nov;261(5 Pt 2):H1617-25). This will help negate the confounding influence of fiber XSA from estimates of Dm within and among subjects and better reflect the existent Dm.

We thank the reviewer for the comment and suggestions. We found great interest in reading the studies suggested. The figure of the work by prof. Hoppeler and colleagues (Respir Physiol. 1981 Apr;44(1):129-50. doi: 10.1016/0034-5687(81)90080-3.) will be on our board for future years.

Fig. 1. Model of respiratory system subdivided into three compartments with O_2 flow rates as function of partial pressure gradients and conductances at each level. The flow rate from the capillaries into the cells [$\dot{V}_{O_2}(B-C)$] depends on a diffusion conductance [\dot{D}_c] which is related to the capillary volume and surface [$V(c)$ and $S(c)$] and to some diffusion distance [δ] from the capillary to the mitochondria.

As reported in the discussion of the original version of our work (“C:F is a structural contributor to DmO_2 , but may not reflect to the true capacity for O_2 diffusion in tissue with variable capillary content of red blood cells (Lines 502)”), we were aware of the limitations of using C:F as proxy of DmO_2 and we agree that number of RBCs flowing in the capillaries adjacent to muscle fibers at any given instant is the best estimator of Dm . Since RBC number is not known in our study, as suggested by the reviewer, capillary volume density is a better alternative than C:F. However, longitudinal muscle sections were not performed as well muscles were not appropriately treated for TEM analyses. Thus, it was not possible to determine tortuosity (γ), capillary volume (V_c) and fiber volume (V_f) measurements in our study. As consequence, we decided to use C:F from immunofluorescence of transverse sections to infer about capillarization of skeletal muscle. Nevertheless, the reviewer is correct that this approach does not take into account the heterogeneity of single fiber cross-section area, which was present in our samples. Thus, we measured fiber cross-sectional area (fiber CSA data are now included in the revised manuscript on line 355) and calculated capillary density (CD) using the following equation:

$$N_A(c, f) = N(c)/A(f)$$

More specifically, total number of capillaries was divided to the associated area of muscle fibers. We are aware this approach has some pitfalls compared to capillary volume density, but it has been already used to estimate DmO_2 (e.g. Hepple et al, J Appl Physiol 2000 Feb;88(2):560-6. doi: 10.1152/jappl.2000.88.2.560.)

CD ranged from 348 to 586 mm^{-2} . The figure below shows no relationship between CD and C:F and it is evident that there were changes in capillarization respect to C:F in the same subjects. For example, S2 and S9 shows an opposite trend in capillarization estimation when CD or C:F are used.

In the revised version of our manuscript, C:F data were replaced by CD and we edited the text to referring to capillary density as an ex-vivo approach to estimate DmO_2 . This major change, however, did not alter the overall findings or conclusions of the study because the data showed that CD and Δk were significantly correlated ($r = -0.68$, $r^2 = 0.46$ and $p = 0.015$).

25 To help the reader reconcile the present measurements with in vivo and other studies, please include measurement of O_2 flux in ml/min/100 g or such - possibly in addition to that presented now (here and throughout).

Thank you for the suggestion. Values in pmol of O_2 were converted in ml O_2 by the gas law ($V = nRT/P$) where V is volume, n is the concentration expressed in mol, R is the gas constant ($R = 0.082$), T is the temperature and P is the pressure. T was set at 37° (310,15 K) according to the value in the respirometry chamber. P values, expressed in atmospheres, were modified according to recorded values during HRR measurements. Data of O_2 were then normalized for 100g of muscle tissue. New data were inserted in the text (lines 27 and 350-352).

28 Why was citrate synthase activity (or such) not measured on the biopsies?

We thank the reviewer for raising this issue. We actually measured CS activity in our samples and data are now reported in the text (line 352). Although O_2 flux and mitochondrial respiration data can be normalized to CS activity to take into account mitochondrial content of the samples, we did not change our figures reporting the correlation between HRR and NIRS to let the readers compare our data with those by Ryan et al (Ryan et al., J Physiol. 2014 Aug 1;592(15):3231-41. doi: 10.1113/jphysiol.2014.274456).

52 Probably cite Richardson et al. 1995 here as well as Clanton.

Reference was added (line 57).

54 Whether capillary-to-mitochondrial diffusion distances in skeletal muscle are of any importance at all has been seriously questioned by the data from Honig and colleagues (e.g., Adv Exp Med Biol. 1986;200:487-94), Hepple et al. (J Appl Physiol (1985). 2000 Feb;88(2):560-6) as well as Richardson and colleagues' proton-MRS work and the development of novel models of capillary function that incorporate the latest findings in this rapidly-developing field (Poole et al. Eur J Appl Physiol. 2022 Jan;122(1):7-28).

Thank you for pointing that out. O_2 diffusion distance was removed (line 60).

58 Please consider adding Roca et al. (1992) in with Richardson et al. here as above.

Reference was added (line 66).

69 For ease of reading, please state 2 unknowns here.

Information was added between brackets (line 77).

87 Impressive though S&G, 1983 is, subsequent work on capillary: fiber surface ratio and especially capillary volume density - as it impacts both capillary RBC transit time and the instantaneous RBC number and Hb volume in approximation with the muscle fibers - represent probably a far more important index of DmO₂ (see above).

Text was modified according to the new approach of estimating DmO₂ by capillary density (lines 95-96).

104 Given the minimal extra work required and the Cores of Reproducibility initiatives on this precise topic it was disappointing that the authors did not validate a VO₂max (J Appl Physiol (1985). 2017 Apr 1;122(4):997-1002). Maximal HR and RER are not adequate indicators of such.

We thank the reviewer for the suggestion, the validation was not performed at this time. We will ensure to include the procedure in future studies.

120 Perhaps I'm being wooden here but, at this location and subsequently, the "0-10% PN does not appear to cohere with either Figure 1 (LOW is 40-50% TSI as indicated) or subsequent data and text. Please clarify and make internally consistent.

Physiological normalization (PN; see also Adami et al., Respir Physiol Neurobiol. 2017 Jan;235:18-26. doi: 10.1016/j.resp.2016.09.008.) was identified during prolonged ischemia and it was comprised between TSI min (deflection point=0%) and TSI max (after reperfusion=100%). Taking PN as 100% range, the LOW range was set from 0 to 10% of PN which corresponded to ~40 to 50% of TSI value for the data obtained in the typical subject shown in figure 1. We modified the figure 1 and the text (lines 177-188) to clarify as requested.

144 Clarify when (i.e., at rest) that the initial value was set to "0" and, perhaps include oxy and deoxy Mb in parentheses. As I suspect Professor Shunsaku Koga would agree TRS would have been better than CWSR-NIRS.

Thank you. The zero setting was obtained at rest, before the prolonged ischemia. This point was clarified in the revised version of the manuscript (lines 158) as well as Mb was included in the acronym throughout the entire manuscript (e.g. lines 156-161). The reviewer is correct, and we would have liked to conduct our experiments with more advanced NIRS equipment. That our proposed technique relies on relative, not absolute changes, and can be made without expensive advanced NIRS equipment, may, we believe, make it more accessible for use by others in the scientific community.

147 Define "SRS" if not already.

Thank you for pointing that out. The definition of SRS was added (line 162).

157 "tHb+tMb"

The change was made (line 159) as requested.

165 As above, not clear what or where 0-10% of PN is.

Please refer to the comment above.

167 Much appreciated that a test/retest was performed.

We thank the reviewer for the comment.

Figure 2 - perhaps simplify the time axis - or does 1550 s have some other significance? Identify

PN point and also consider switching panels C and D so they line up below the appropriate TSI graphs.

The time values on x axis were selected as those reported in Figure 1 to help the readers to understand that data analyses were performed starting from TSI value obtained during the experimental protocol shown in Figure 1.

As for panel labeling, we realized there was an error in the caption. Thank you for pointing that out. Panel C and D report mono-exponential fitting of TSI data obtained in LOW and HIGH, respectively. Text in caption was corrected for figure 2 (207-208).

186 Only 1 grey point I could see.

Thank you for pointing that out. Text was corrected (line 208).

218 Please present chamber PO₂ as well as concentration to allow reader to appreciate how these may relate to physiologic values (or not!).

We added PO₂ values corresponding to chamber O₂ concentration (lines 240-241)

234 As above. Please also put these in ml O₂/min/100 g or kg.

Values in ml O₂/min/100 g were added in the text (lines 350-352).

249 More details of morphometry needed as well as measurement of fibers XSA, capillary density and estimate of capillary volume density.

As previously stated, data from muscle samples were re-analyzed to calculate muscle fiber CSA and capillary density. The text was modified accordingly (Lines 95-96, 278-283, and 297-300)

272 Unclear. Was PN then ~125%? Please clarify as requested above.

Please refer to comment to line 120.

275 The lack of significant difference of mean values between two tests just might be evidence for no overarching effect of a "practice" run. Please present the coefficient of variation to assess better 'reproducibility.'

Thank you for the suggestion. CV data were included in the revised version of the manuscript (Lines 329-330)

279 A correlation of 0.67 really only explains 45% of the variability. What factors might contribute to the "other 55%"?

This is an interesting point. It has been suggested that poor reproducibility can be explained principally by a small increase in mVO₂ and only modest deoxygenation during contractions (Adami et al., *Respir Physiol Neurobiol.* 2017 Jan;235:18-26. doi: 10.1016/j.resp.2016.09.008). In our case, we tried to exclude this factor by performing NIRS protocol immediately after a constant work rate exercise, which should have yielded sufficient muscle activation. This was effectively the case because TSI values were $50.4 \pm 6.4\%$ of PN. Thus, we can reasonably exclude that insufficient contractile stimulus for mitochondrial activation occurred. However, it is well known that NIRS signal can be affected by adipose tissue so cannot exclude large adipose layer may have contributed to poor test quality.

282 "tHb+tMb" was 3.14 micromole - wasn't this set to "0"? please explain.

The reviewer is correct. Relative concentrations of NIRS parameters were measured with respect to an initial value obtained at rest arbitrarily set equal to zero. This procedure was performed at the start of the experiments, before any other procedure. We added this information in the methods (lines 158). The values reported in this paragraph (line 335) are referred to those obtained between the prolonged ischemia and the first cycling exercise. In our opinion, the release of vasoactive

substances during ischemia may have caused vasodilation of microcirculation and can explain the change in tot(Hb+Mb) from baseline. (McLay et al., *Exp Physiol* 101: 34–40, 2016. doi:10.1113/EP085406. ; McLay et al., *Exp Physiol* 101: 1309–1318, 2016. doi:10.1113/EP085843. ; Barstow, *J Appl Physiol* (1985). 2019 May 1;126(5):1360-1376. doi: 10.1152/jappphysiol.00166.2018). The symbol has been modified to reflect this change: $\Delta[\text{tot}(\text{Hb}+\text{Mb})]$

284 How much Hb volume does the ~6 micromoles of elevated [Hb] account for? This is a critical calculation because, combined with the suggested estimation of capillary volume density, it allows the mean impact of exercise and occlusion on capillary hematocrit - and therefore DmO_2 - to be estimated. For example, supposing that the capillary density is 300 mm⁻¹ (allowing for capillary tortuosity and branching and a mean capillary diameter of 6 microns, capillary volume density would be 1.1%) and Hcrit were presumed to be ~30%. It can be calculated that the 6 micromole (0.39 g Hb per L) increase in [Hb] would elevate capillary Hcrit from 30-34% which is, I think, quite reasonable, and adds a practical and logical quantitative coherence to the data interpretation. Perhaps enlisting a muscle morphometry specialist to help with this?

We thank the reviewer for this interesting suggestion, which required us to carefully read the literature and make some new calculations. We have summarized our reasoning in the following paragraphs. To follow these calculations, please take into account we assumed that:

- NIRS data are referred to a definite volume of tissue;
- capillary blood volume was 1.3% of tissue;
- Hct in the capillaries was 15%, which should correspond to $1.76 \cdot 10^{12}$ RBC and 5.29 g/dL of [hemoglobin] in the capillaries (Poole et al., *Comp Biochem Physiol A Mol Integr Physiol.* 2021 Mar;253:110852. doi: 10.1016/j.cbpa.2020.110852.)

Using this, an ~10 μM change in $\Delta[\text{tot}(\text{Hb}+\text{Mb})]$ from rest to NIRS protocol (exercise + occlusions), would suggest an increase of $2.15 \cdot 10^{12}$ RBC and 6.45 g/dL of [hemoglobin] in muscle capillaries. As a consequence, Hct may have increased to ~18%, leading to a total Hct of 33%. These estimations agree with the calculations suggested by the reviewer, and we confirm that our NIRS data seems to suggest a modest increase in capillary hematocrit from resting conditions to NIRS protocol and also change in RBCs number (a good estimator of DmO_2) between HIGH and LOW is negligible.

Figure 3. Please state c.v.

CV was added in the text (line 332) and to Figure 3 as requested.

Figure 5. What is the correlation between k_{HIGH} and C:F (or better - capillary volume density.

We thank the reviewer for this question. k_{HIGH} and CD were not significantly correlated ($r^2=0.20$, $p=0.14$). This reinforces idea that the recovery rate constant of mVO_2 is not limited by O_2 availability when TSI is high. Any particular rate of $\text{mVO}_{2\text{max}}$ (or maximal enzymatic activity) can be moderated by both O_2 convection and diffusion. Therefore, although we might anticipate a significant association between maximal respiratory capacity (here represented by k_{HIGH}) and CD, these are independent variables that do not, by necessity, have to be strongly related. Nevertheless, a close relationship between O_2 delivery to the muscles and mitochondrial volume density has been reported in both rodents and humans (see e.g. Hoppeler and Kayar, Capillarity and oxidative capacity of muscles. *NIPS* 3:113–116, 1988; Mathieu-Costello, Geometry of blood-tissue exchange in bat flight muscle compared with bat hindlimb and rat soleus muscle. *Am. J. Physiol.* 262:R955–R965, 1992). Further work will be needed to explore these associations in vivo.

327 and 335 As above....better for cap vol density.

As previously stated (see comment for line 21), C:F was replaced by CD to estimate DmO_2 .

Unfortunately, we were not able to measure capillary volume density, which would have been a more accurate approach to estimate DmO_2 .

330 The implications of this work, as several of the authors have stated in previous investigation, extends crucially to heart failure. That point could be made usefully here.

We thank the reviewer for the suggestion. A sentence was added in the manuscript to highlight the useful perspective of the novel approach described in the present study (Lines 400-402)

343 "function" is meant surely?

Text was corrected (line 407).

394 Perhaps discuss briefly the possible concerns with hyperoxic respirometry?

Thank you for the suggestion. High-resolution respirometry measurements were conducted between 250 to 450 $nmol \cdot ml^{-1}$ (corresponding to a PO_2 around 250mmHg). Consequently, ex vivo mitochondrial function was measured in non-limiting O_2 condition. We add a sentence to clarify this issue (lines 451).

413 "agnostic" is there a better word/phrase?

The text was changed to "unrelated" (line 486)

415-6 Seems circular logic.

Text was edited to be clearer (lines 487-489).

418-25 The heart presents a good exemplar of some of the problems, discussed above, with using C:F ratio as an index of capillarity (and therefore estimate of Dm). Myocardial C:F is ~ 1.1 and yet, because of the very small myocardial fiber XSA myocardial capillary density and volume density, and therefore, Dm are extremely high.

Thank you for the comment and the enlightening insights. Capillary density and volume density are surely a better parameter correlated with DmO_2 .

433 This seems a bit harsh on the great Peter Wagner whose relevant experiments were, inarguably, groundbreaking and moved this field forward considerably.

Text was modified to improve clarity (lines 506-509).

445-7 As above provide quantitative reasoning.

Quantitative data were added to the text (lines 520-521).

449 "more advanced" might be qualified to "as yet undeveloped".

Text was rephrased as suggested (line 526).

456 Please state relative volumes for biopsy vs NIRS.

Muscle biopsy sample were $\sim 100mg$ which should correspond to a volume of $\sim 0.94mm^3$, assuming a muscle tissue density of $\sim 1.06 kg/l$. On the other hand, NIRS should investigate a muscle volume of $\sim 2-3 cm^3$, assuming an inter-optode distance of 3-4 cm and an amount of tissue approximately 1.5 – 2 cm beneath the probe (0.5-1 cm of adipose tissue thickness). Text was corrected (line 534-535).

464 What effect does only recruiting a modest portion of the muscle interrogated have on the measured kinetics/absolute levels measured?

VO_2 kinetics are mathematically independent of absolute VO_2 . Therefore, if only even a very small portion of the muscle under the NIRS probe were activated, with the remaining muscle respiring at

a basal state, we would still (assuming good sensitivity of NIRS) be able to measure k from the activated fibers, which is associated with $m\dot{V}O_{2\max}$. Under such a condition, with a small portion of the interrogated muscle recruited, the absolute $\dot{V}O_2$ might be low. In addition, relatively superficial muscle contains a greater fraction of fast, glycolytic fibers. Clearly some of the fibers under the NIRS probe were activated (the exercise increased $m\dot{V}O_2$). Given the moderate intensity of the exercise, and assuming sequential motor-unit recruitment, the activated fibers were “visible” by NIRS, and therefore k , would likely be biased towards lower order fibers (slow, oxidative). The respiratory capacity in the fiber bundles during HRR, on the other hand, is representative of all the fibers in the bundle, not only those activated by exercise during the NIRS protocol. These differences might explain some of the remaining unexplained variance between methods. These points have been added to the discussion (lines 537-539).

469 Perhaps "validity" assessment should be left to the reader: "reasonable" would be better here?
Text was rephrased as suggested (line 548).

Referee #2:

General Comments

This study examined if the recovery rate constant of mVO₂ in high tissue saturation index (TSI) is associated with muscle oxidative capacity and if the difference in the recovery rate constant of mVO₂ between non-O₂ limiting (HIGH) and O₂ limiting (LOW) TSI conditions. This study has been well-designed and elaborately conducted and contains invaluable outcomes. However, some of the theoretical basis should be thoroughly explained for readers to understand.

We thank the reviewer for the time dedicated in reviewing our manuscript. We are happy to know that our study raised your interest. We modified the manuscript according to your suggestions and we believe the current version is improved.

Major Comments

The authors thought that they would be able to solve for diffusion capacity of O₂ (DmO₂) by simultaneous subtraction of the two unknowns in Eq. 1. It is true that intramyocyte PO₂ (PimO₂) should be almost constant during moderate- to high-intensity of exercise. However it is evident that microvascular PO₂ (PmvO₂) in HIGH and LOW conditions in this study is not equivalent as shown in Figure 2 (although TSI is not a direct measure of PO₂). The authors should explain this issue, for example, by showing an equation of this estimation.

Thank you for the comment. The NIRS-based protocol has been set with the aim to investigate two different O₂ availability conditions (HIGH vs LOW), assuming that this approach would have given us the possibility to measure of muscle oxygen consumption (mVO₂) at different PmvO₂. We used recovery rate constant (*k*) as proxy for mVO₂ and TSI as a proxy of PmvO₂. Although not without limitations, this approach led us to solve the Fick's law of diffusion as follows:

$$k = DmO_2 \times TSI$$

where PimO₂ is considered negligible.

Then, we calculated *k* for both high (*k*_{HIGH}) and low (*k*_{LOW}) TSI values and we estimated DmO₂ from the difference in *k* values obtained in these two conditions. We assumed that mV'O₂ was the same in both conditions (the exercise task was the same). But during recovery from exercise we manipulated O₂ delivery and observed the effect in *k*. This allowed us to calculate Δ*k* and ΔTSI, which were the difference between the HIGH and LOW conditions. We then tested the hypothesis that solving this equation would reflect DmO₂, by comparison with capillary density (CD)

$$\Delta k / \Delta TSI = DmO_2 = CD$$

We are very careful to experimentally control the ΔTSI for each participant, using from 0 to 10% of PN for LOW and from 50 to 60% of PN for HIGH (Figure 1). Since no absolute values were present, and the values were standardized among subjects, ΔTSI can be neglected. We have modified the text to help the readers follow our reasoning (lines 77-78).

The authors should discuss the NIR signal contribution from Hb and Mb and the influence of the Mb signal contribution on the assumption used in this study.

Thank you for pointing that out. The NIRS signals are the result of O₂ saturations of both the heme groups of Hb in the blood vessels (predominantly from the capillaries) and the heme group of Mb in muscle fibers. Therefore, NIRS reflects changes in capillary (Hb-related) and intracellular (Mb-related) O₂ levels. In our study, Hb was assumed as responsible for most of the overall NIRS signal

changes observed according to the correlation between muscle oxygenation values obtained by NIRS and O₂ saturation in venous blood (e.g.: Seiyama et al. J Biochem. 1988 Mar;103(3):419-24. doi: 10.1093/oxfordjournals.jbchem.a122285). Nevertheless, this approach does not consider the influence of Mb signal which was assumed to be of minor impact compared to the contribution of Hb. We modified the Methods section to clarify this aspect (Lines 149-155) and included “Mb” term in all the acronym related to NIRS variables.

It seems that the authors did not mention a randomization of LOW and HIGH conditions. Are there any "order effect" on LOW and HIGH conditions. It is probable that the 1st trial of exercise would facilitate the oxidative capacity in the 2nd trial.

Thank you for pointing that out. In our study, the LOW and HIGH conditions were randomized (lines 189-190). We rephrased the text to clarify this point in our work and avoid confusion.

As the correlation between JO₂ and k-LOW is insignificant and rather "flat", the significant correlation between C:F and delta-k is primarily derived from the correlation of JO₂ to k-HIGH. How about the correlation between C:F and k-HIGH? It is not surprising to find a good correlation between C:F and k-HIGH because VO_{2max} and capillarity correlates well.

We thank the reviewer for the interesting suggestion, which was also raised by reviewer 1. k_{HIGH} and CD were not significantly correlated ($r^2=0.20$, $p=0.14$). This reinforces idea that the recovery rate constant of mVO₂ is not limited by O₂ availability when TSI is high. Any particular rate of mVO_{2max} (or maximal enzymatic activity) can be moderated by both O₂ convection and diffusion. Therefore, although we might anticipate a significant association between maximal respiratory capacity (here represented by k_{HIGH}) and CD, these are independent variables that do not, by necessity, have to be strongly related. Indeed, CD does not take account the degree of capillary tortuosity, which is well-known to introduce considerable variability into measurements of O₂ delivery to muscle fibers (e.g., Mathieu-Costello O. Microvasc Res. 1987 Jan;33(1):98-117. doi: 10.1016/0026-2862(87)90010-0). A classic example can be found in the work by Prof. Poole and Mathieu-Costello (Microcirculation. 1996 Jun;3(2):175-86. doi: 10.3109/10739689609148286), where capillary surface per fiber surface was significantly correlated with mitochondrial volume per unit fiber length for both soleus and plantaris muscles but capillary-to-fiber ratio and either citrate synthase activity or mitochondrial volume per fiber length were not correlated in plantaris muscles. Nevertheless, a close relationship between O₂ delivery to the muscles and mitochondrial volume density has been reported in both rodents and humans (see e.g. Hoppeler and Kayar, Capillarity and oxidative capacity of muscles. NIPS 3:113–116, 1988; Mathieu-Costello, Geometry of blood-tissue exchange in bat flight muscle compared with bat hindlimb and rat soleus muscle. Am. J. Physiol. 262:R955–R965, 1992). Further work will be needed to explore these associations in vivo.

The authors should discuss a significance of the O₂ level in LOW (0-10%) and HIGH (50-60%). Why this level was used.

According to Adami and Rossiter (J Appl Physiol (1985). 2018 Jan 1;124(1):245-248. doi: 10.1152/jappphysiol.00445.2017) NIRS-based protocol relies on two main assumptions: occlusions must be performed in condition of sufficient mitochondrial activation and, in order that k be a valid estimate muscle oxidative capacity, occlusions must be performed in condition of non-limiting O₂ availability (i.e. maintaining TSI above 50% of the physiological normalization). Low PO₂ slows the recovery of muscle oxygen consumption (J Appl Physiol (1985). 2004 Sep;97(3):1077-81. doi: 10.1152/jappphysiol.01321.2003).

As such, we utilized a range higher than 50% for non-O₂ limiting condition (50-60% PN for HIGH), whereas 0-10% of PN was selected for LOW range to be sure to investigate a condition of poorly-oxygenated muscle but, at the same time, not overstepping the deflection point where TSI signal during occlusions lose linearity (lines 183-188).

In figure 3, it is suggested that LOW and HIGH plots be distinct by using a different plot design such as filled and unfilled circles.

The suggested change was made (line 341).

Minor Comments

L 112: VISIT 3 should be the 2nd TRIAL.

Figure 1 was corrected as suggested (line 121).

L 274: need (,) before respectively.

Text was corrected.

END OF COMMENTS

The Physiological Society is a company limited by guarantee. Registered in England and Wales, No. 00323575. Registered Office: Hodgkin Huxley House, 30 Farringdon Lane, London, EC1R 3AW, UK. Registered Charity No. 211585. The Physiological Society and The Journal of Physiology are registered trademarks.

This email and any files transmitted with it are confidential and intended solely for the use of the individual or entity to whom they are addressed. If you have received this email in error please notify the sender. If you are not the named addressee you should not disseminate, distribute or copy this e-mail. The Physiological Society may monitor email traffic data.

The Physiological Society has taken reasonable precautions to ensure no viruses are present in this email, however does not accept responsibility for any loss or damage arising from the use of this email or attachments.

Dear Dr Porcelli,

Re: JP-RP-2022-283267R1 "Near-infrared spectroscopy estimation of combined skeletal muscle oxidative capacity and O₂ diffusion capacity in humans" by Andrea Pilotto, Alessandra Adami, Raffaele Mazzolari, Lorenza Brocca, Emanuela Crea, Lucrezia Zuccarelli, Maria Antonietta Pellegrino, Roberto Bottinelli, Bruno Grassi, Harry B Rossiter, and Simone Porcelli

Thank you for submitting your manuscript to The Journal of Physiology. It has been assessed by a Reviewing Editor and by 2 expert Referees and I am pleased to tell you that it is considered to be acceptable for publication following satisfactory revision.

The reports are copied at the end of this email. Please address all of the points and incorporate all requested revisions, or explain in your Response to Referees why a change has not been made.

NEW POLICY: In order to improve the transparency of its peer review process The Journal of Physiology publishes online as supporting information the peer review history of all articles accepted for publication. Readers will have access to decision letters, including all Editors' comments and referee reports, for each version of the manuscript and any author responses to peer review comments. Referees can decide whether or not they wish to be named on the peer review history document.

Authors are asked to use The Journal's premium BioRender (<https://biorender.com/>) account to create/redraw their Abstract Figures. Information on how to access The Journal's premium BioRender account is here:

<https://physoc.onlinelibrary.wiley.com/journal/14697793/biorender-access> and authors are expected to use this service. This will enable Authors to download high-resolution versions of their figures. The link provided should only be used for the purposes of this submission. Authors will be charged for figures created on this premium BioRender account if they are not related to this manuscript submission.

I hope you will find the comments helpful and have no difficulty returning your revisions within 4 weeks.

Your revised manuscript should be submitted online using the links in Author Tasks: Link Not Available.

Any image files uploaded with the previous version are retained on the system. Please ensure you replace or remove all files that have been revised.

REVISION CHECKLIST:

- Article file, including any tables and figure legends, must be in an editable format (eg Word)
- Abstract figure file (see above)
- Statistical Summary Document
- Upload each figure as a separate high quality file
- Upload a full Response to Referees, including a response to any Senior and Reviewing Editor Comments;
- Upload a copy of the manuscript with the changes highlighted.

- A potential 'Cover Art' file for consideration as the Issue's cover image;
- Appropriate Supporting Information (Video, audio or data set https://jp.msubmit.net/cgi-bin/main.plex?form_type=display_requirements#supp).

To create your 'Response to Referees' copy all the reports, including any comments from the Senior and Reviewing Editors, into a Word, or similar, file and respond to each point in colour or CAPITALS and upload this when you submit your revision.

I look forward to receiving your revised submission.

If you have any queries please reply to this email and staff will be happy to assist.

Yours sincerely,

Michael C. Hogan
Senior Editor
The Journal of Physiology
<https://jp.msubmit.net>
<http://jp.physoc.org>
The Physiological Society
Hodgkin Huxley House
30 Farringdon Lane
London, EC1R 3AW
UK
<http://www.physoc.org>
<http://journals.physoc.org>

REQUIRED ITEMS:

- The contact information provided for the person responsible for 'Research Governance' at your institution is an author on this paper. Please provide an alternative contact who is not an author on this paper or confirm that the author whose email was provided has sole responsibility for research governance. This is the person who is responsible for regulations, principles and standards of good practice in research carried out at the institution, for instance the ethical treatment of animals, the keeping of proper experimental records or the reporting of results.

- The Journal of Physiology funds authors of provisionally accepted papers to use the premium BioRender site to create high resolution schematic figures. Follow this link and enter your details and the manuscript number to create and download figures. Upload these as the figure files for your revised submission. If you choose not to take up this offer we require figures to be of similar quality and resolution. If you are opting out of this service to authors, state this in the Comments section on the Detailed Information page of the submission form. The link provided should only be used for the purposes of this submission. Authors will be charged for figures created on this premium BioRender account if they are not related to this manuscript submission.

- Please ensure that any tables are in Word format and are, wherever possible, embedded in the article file itself.

EDITOR COMMENTS

Reviewing Editor:

Thank you for addressing the comments made by the reviewers both of who have highlighted the quality and likely impact of the work performed. Prior to the article being suitable for publication we would however ask you to address the comment from reviewer 2 - you have provided adequate justification to the reviewer but the manuscript would benefit from the general content of this statement being included for clarity.

In the column Experiential question number the numbers included do not match up with the stated hypothesis. Could you include a few words in this column for each test performed stating the question being answered?

Minor comments:

In either the table 1 or its legend, could you indicate the number of participants studied?

Could you modify the statistical summary document so that the experimental question number column directly relates to one of the 2 stated hypothesis (or add more hypothesis), or include a short statement in the column indicating the experimental question being asked?

As part of the submission checklist you are required to provide contact details for the person responsible for Research Governance at the institution where the research was carried out. Please ensure this is completed.

REFEREE COMMENTS

Referee #1:

The authors have performed an excellent revision that has substantially improved the correctness, clarity and presentation

of these data. The matching of structure and function as performed herein adds considerably to our understanding of muscle oxidative function.

Referee #2:

All the raised comments and suggestions have been appropriately dealt with and modified in the text except for the following issue.

The reviewer strongly recommends including the essence of the following statement in the text.

Thank you for the comment. The NIRS-based protocol has been set with the aim to investigate two different O₂ availability conditions (HIGH vs LOW), assuming that this approach would have given us the possibility to measure of muscle oxygen consumption (mVO₂) at different PmvO₂. We used recovery rate constant (k) as proxy for mVO₂ and TSI as a proxy of PmvO₂. Although not without limitations, this approach led us to solve the Fick's law of diffusion as follows:

$$k = DmO_2 \times TSI$$

where PimO₂ is considered negligible.

Then, we calculated k for both high (kHIGH) and low (kLOW) TSI values and we estimated DmO₂ from the difference in k values obtained in these two conditions. We assumed that mV'O₂ was the same in both conditions (the exercise task was the same). But during recovery from exercise we manipulated O₂ delivery and observed the effect in k. This allowed us to calculate Δk and ΔTSI , which were the difference between the HIGH and LOW conditions. We then tested the hypothesis that solving this equation would reflect DmO₂, by comparison with capillary density (CD).

$$\Delta k / \Delta TSI = DmO_2 = CD$$

We are very careful to experimentally control the ΔTSI for each participant, using from 0 to 10% of PN for LOW and from 50 to 60% of PN for HIGH (Figure 1). Since no absolute values were present, and the values were standardized among subjects, ΔTSI can be neglected. We have modified the text to help the readers follow our reasoning (lines 77-78).

END OF COMMENTS

1st Confidential Review

04-Jul-2022

EDITOR COMMENTS

Reviewing Editor:

Thank you for addressing the comments made by the reviewers both of who have highlighted the quality and likely impact of the work performed. Prior to the article being suitable for publication we would however ask you to address the comment from reviewer 2 - you have provided adequate justification to the reviewer but the manuscript would benefit from the general content of this statement being included for clarity.

Dear Editor, we are happy to know that our revision was appreciated, and we revised the manuscript according to last suggestions.

A detailed list of responses to the editor and reviewers' suggestions follows. Line numbers refer to the red lined version of the manuscript, when needed.

In the column "Experiential question number" the numbers included do not match up with the stated hypothesis. Could you include a few words in this column for each test performed stating the question being answered?

According to the request, we specified the experimental question and included information about one of the three hypotheses tested

Minor comments:

In either the table 1 or its legend, could you indicate the number of participants studied?

We are sorry for that. Information were added to the legend

Could you modify the statistical summary document so that the experimental question number column directly relates to one of the 2 stated hypothesis (or add more hypothesis), or include a short statement in the column indicating the experimental question being asked?

As reported above, we specified the experimental question and included information about one of the three hypotheses tested

As part of the submission checklist you are required to provide contact details for the person responsible for Research Governance at the institution where the research was carried out. Please ensure this is completed.

The change was made

Referee #1:

The authors have performed an excellent revision that has substantially improved the correctness, clarity and presentation of these data. The matching of structure and function as performed herein adds considerably to our understanding of muscle oxidative function.

We thank the reviewer for the comments and we are happy to know that our study raised his/her interest.

Referee #2:

All the raised comments and suggestions have been appropriately dealt with and modified in the text except for the following issue.

The reviewer strongly recommends including the essence of the following statement in the text.

We thank the reviewer for the time dedicated in reviewing our manuscript. We modified the introduction according his/her suggestion (lines 84-104).

Dear Dr Porcelli,

Re: JP-RP-2022-283267R2 "Near-infrared spectroscopy estimation of combined skeletal muscle oxidative capacity and O₂ diffusion capacity in humans" by Andrea Pilotto, Alessandra Adami, Raffaele Mazzolari, Lorenza Brocca, Emanuela Crea, Lucrezia Zuccarelli, Maria Antonietta Pellegrino, Roberto Bottinelli, Bruno Grassi, Harry B Rossiter, and Simone Porcelli

I am pleased to tell you that your paper has been accepted for publication in The Journal of Physiology.

NEW POLICY: In order to improve the transparency of its peer review process The Journal of Physiology publishes online as supporting information the peer review history of all articles accepted for publication. Readers will have access to decision letters, including all Editors' comments and referee reports, for each version of the manuscript and any author responses to peer review comments. Referees can decide whether or not they wish to be named on the peer review history document.

The last Word version of the paper submitted will be used by the Production Editors to prepare your proof. When this is ready you will receive an email containing a link to Wiley's Online Proofing System. The proof should be checked and corrected as quickly as possible.

Authors should note that it is too late at this point to offer corrections prior to proofing. The accepted version will be published online, ahead of the copy edited and typeset version being made available. Major corrections at proof stage, such as changes to figures, will be referred to the Reviewing Editor for approval before they can be incorporated. Only minor changes, such as to style and consistency, should be made a proof stage. Changes that need to be made after proof stage will usually require a formal correction notice.

All queries at proof stage should be sent to TJP@wiley.com.

Are you on Twitter? Once your paper is online, why not share your achievement with your followers. Please tag The Journal (@jphysiol) in any tweets and we will share your accepted paper with our 23,000+ followers!

Yours sincerely,

Michael C. Hogan
Senior Editor
The Journal of Physiology
<https://jp.msubmit.net>
<http://jp.physoc.org>
The Physiological Society
Hodgkin Huxley House
30 Farringdon Lane
London, EC1R 3AW
UK
<http://www.physoc.org>
<http://journals.physoc.org>

P.S. - You can help your research get the attention it deserves! Check out Wiley's free Promotion Guide for best-practice recommendations for promoting your work at www.wileyauthors.com/eeo/guide. And learn more about Wiley Editing Services which offers professional video, design, and writing services to create shareable video abstracts, infographics, conference posters, lay summaries, and research news stories for your research at www.wileyauthors.com/eeo/promotion.

*** IMPORTANT NOTICE ABOUT OPEN ACCESS ***

To assist authors whose funding agencies mandate public access to published research findings sooner than 12 months after publication The Journal of Physiology allows authors to pay an open access (OA) fee to have their papers made freely available immediately on publication.

You will receive an email from Wiley with details on how to register or log-in to Wiley Authors Services where you will be able to place an OnlineOpen order.

You can check if your funder or institution has a Wiley Open Access Account here: <https://authorservices.wiley.com/author-resources/Journal-Authors/licensing-and-open-access/open-access/author-compliance-tool.html>.

Your article will be made Open Access upon publication, or as soon as payment is received.

If you wish to put your paper on an OA website such as PMC or UKPMC or your institutional repository within 12 months of publication you must pay the open access fee, which covers the cost of publication.

OnlineOpen articles are deposited in PubMed Central (PMC) and PMC mirror sites. Authors of OnlineOpen articles are permitted to post the final, published PDF of their article on a website, institutional repository, or other free public server, immediately on publication.

Note to NIH-funded authors: The Journal of Physiology is published on PMC 12 months after publication, NIH-funded authors DO NOT NEED to pay to publish and DO NOT NEED to post their accepted papers on PMC.

EDITOR COMMENTS

Thank you for addressing the comments made by the reviewers.

2nd Confidential Review

15-Jul-2022